# Memory-Free Continual Learning with Null Space Adaptation for Zero-Shot Vision-Language Models

**Yujin Jo and Taesup Kim**[†]

Graduate School of Data Science
Seoul National University

## Abstract

Pre-trained vision-language models (VLMs), such as CLIP, have demonstrated remarkable zero-shot generalization, enabling deployment in a wide range of real-world tasks without additional task-specific training. However, in real deployment scenarios with evolving environments or emerging classes, these models inevitably face distributional shifts and novel tasks. In such contexts, static zero-shot capabilities are insufficient, and there is a growing need for continual learning methods that allow models to adapt over time while avoiding catastrophic forgetting. We introduce NuSA-CL (Null Space Adaptation for Continual Learning), a lightweight memory-free continual learning framework designed to address this challenge. NuSA-CL employs low-rank adaptation and constrains task-specific weight updates to lie within an approximate null space of the model's current parameters. This strategy minimizes interference with previously acquired knowledge, effectively preserving the zero-shot capabilities of the original model. Unlike methods relying on replay buffers or costly distillation, NuSA-CL imposes minimal computational and memory overhead, making it practical for deployment in resource-constrained, real-world continual learning environments. Experiments show that our framework not only effectively preserves zero-shot transfer capabilities but also achieves highly competitive performance on continual learning benchmarks. These results position NuSA-CL as a practical and scalable solution for continually evolving zero-shot VLMs in real-world applications.

## 1 Introduction

Vision-language foundation models such as CLIP (Radford et al., 2021) have brought about a major shift in artificial intelligence by enabling zero-shot generalization. Their powerful text-image aligned representations now serve as the perceptual core for a new generation of systems, including Multimodal large language models (MLLMs) like LLaVA (Liu et al., 2023; 2024) and Vision-language-action (VLA) models for robotics (Kim et al., 2024; Shukor et al., 2025). However, these advanced systems inherit a critical limitation from their static backbones: In settings where data distributions and user requirements are constantly evolving, their knowledge is frozen. To bridge the gap between static foundation models and the demands of real-world deployment without resorting to massive retraining, Continual Learning (CL) has emerged as a promising solution. CL allows models to incrementally acquire new knowledge while preventing catastrophic forgetting of both pre-trained and previously learned tasks.

Existing CL paradigms, however, face a fundamental scalability wall. Storage-based methods, which rely on experience replay or reference data (Saha et al., 2021; Wang et al., 2021), are inherently constrained by storage costs that grow linearly with the number of tasks. On the other hand, expansion-based methods introduce new modules for each task, such as adapters or prompts (Yu et al., 2024; Tang et al., 2024), forcing a model's parameters and architectural complexity to grow unbounded over time. While effective on short-term benchmarks, these dominant approaches are ill-suited for

---

[†]Corresponding author.

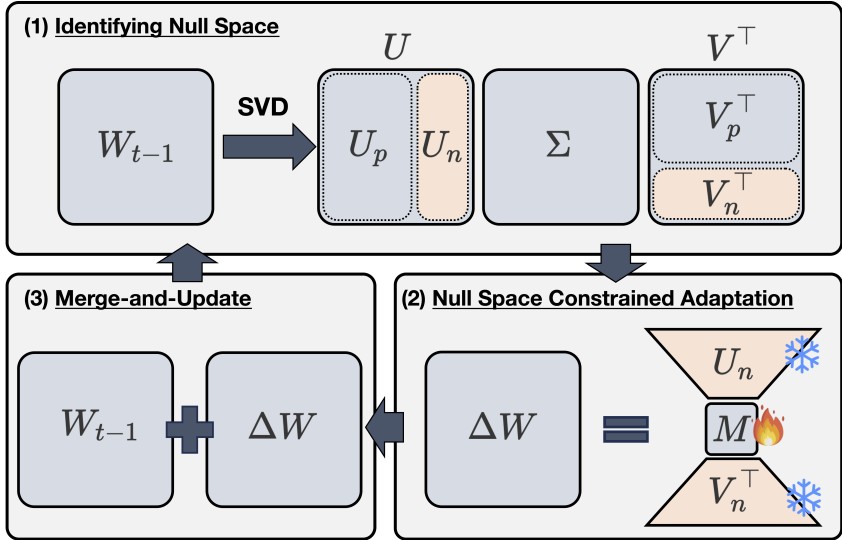

Figure 1: **The NuSA-CL framework.** Starting with the weights from the previous task $W_{t-1}$, we first perform SVD to identify the intrinsic null space. A new low-rank update $\Delta W_t$ is then learned under a persistent constraint that confines it to this space. Finally, the update is merged to produce the new weights $W_t \leftarrow W_{t-1} + \Delta W_t$, completing the cycle.

true lifelong learning. Even many parameter-efficient fine-tuning (PEFT) techniques still depend on explicitly storing prior task information to mitigate interference (Liang & Li, 2024; Lu et al., 2024).

We argue that overcoming this scalability wall requires a paradigm shift from relying on external resources to enabling a model to adapt using only its intrinsic structure. We propose NuSA-CL (Null Space Adaptation for Continual Learning), a continual learning framework that enables a model with a fixed capacity to efficiently reorganize its own knowledge to accommodate new information. NuSA-CL dynamically identifies an underutilized null space in the model's current weights via SVD before each new task and strictly confines all weight updates within these interference-free dimensions throughout training. This data-agnostic process concludes by merging the update into the backbone, maintaining a fixed parameter budget. By preserving the model's core knowledge, NuSA-CL enables stable continual adaptation, offering the ultimate form of scalability with zero storage overhead, zero auxiliary model load, and zero parameter growth which is a crucial set of properties for resource-constrained environments such as autonomous agents or on-device AI.

Our contributions are summarized as follows:

- We propose NuSA-CL, a novel memory-free and resource-efficient continual learning method for vision-language foundation models, designed to operate effectively in resource-constrained environments without relying on memory buffers or knowledge distillation.

- NuSA-CL introduces a null space-constrained low-rank update strategy that integrates new knowledge into an approximate null space of the pre-trained parameters, thereby preserving zero-shot generalization while enabling stable and incremental learning.

- Our method demonstrates significant computational and memory efficiency, making it well-suited for real-world applications that demand lifelong adaptability under limited resources.

## 2 RELATED WORK

### 2.1 CONTINUAL LEARNING WITH PARAMETER-EFFICIENT FINE-TUNING

Continual learning (CL) aims to adapt models to a sequence of tasks without the catastrophic forgetting of previously acquired knowledge (Li & Hoiem, 2017; Rebuffi et al., 2017; Chaudhry

et al., 2019; Ding et al., 2022). A central challenge is the *stability-plasticity trade-off*, which is amplified in foundation models like CLIP, where forgetting undermines not only past tasks but also the general-purpose zero-shot capabilities acquired during pre-training (Wortsman et al., 2022; Zhang et al., 2023; Tan et al., 2024). While full fine-tuning methods like ZSCL (Zheng et al., 2023) can be effective, they often require resource-intensive techniques, hindering their scalability. Parameter-Efficient Fine-Tuning (PEFT) offers a lightweight alternative by restricting updates to a small subset of parameters. This family includes prompt-based methods that isolate task-specific knowledge (Wang et al., 2022b;a) and adapter-based methods that insert small, trainable modules for each task (Yu et al., 2024; Tang et al., 2024; Wu et al., 2025; Wei et al., 2024). However, many of these approaches still externalize new knowledge into task-specific modules, which contributes to parameter growth over long task sequences. In contrast, our work focuses on adapting the model's core weights within a fixed parameter budget.

Our method targets the underlying zero-shot vision–language model itself. Such encoders are increasingly used as multimodal foundation components that extract visual features for larger systems, including MLLMs and VLA agents, so enabling them to continually absorb new visual knowledge under a fixed parameter budget and without external storage is crucial. While higher-level components (e.g., language heads, instruction-tuned decoders, or policy layers) can be continually trained via instruction tuning or task-specific adaptation, our method provides a complementary mechanism at the feature-encoding level rather than a direct competitor to full multimodal generation stacks.

## 2.2 ORTHOGONAL PROJECTION AND NULL SPACE APPROACHES

To explicitly mitigate interference, orthogonal projection techniques constrain parameter updates to subspaces that are orthogonal to those encoding prior knowledge. Prior works typically project new updates away from subspaces identified using stored information, such as past data, features, or gradients (Wang et al., 2021; Saha et al., 2021; Zhao et al., 2023). For instance, InfLoRA (Liang & Li, 2024) adapts this concept to LoRA but still necessitates a memory bank of past gradients to enforce orthogonality. This reliance on external memory contrasts with our strictly memory-free approach. Our method is distinct in that it derives the approximate null space intrinsically from the model's current weight structure via SVD, requiring no access to or storage of past data, features, or gradients.

## 2.3 SVD-GUIDED ADAPTATION IN FOUNDATION MODELS

Several recent methods explore using the spectral properties of a model's weights to guide adaptation, primarily for single-task fine-tuning (Lingam et al., 2024; Yang et al., 2025; Tang et al., 2025). These approaches involve adapting either principal components (Meng et al., 2024) or, conversely, minor low-energy components to minimize interference with pre-trained knowledge (Wang et al., 2025). While insightful, these methods differ from our work in two fundamental ways. First, they are designed for single-task adaptation, not the long-term, sequential learning required in CL. Second, and most critically, they use the low-energy subspace only for initialization, allowing the weight updates to deviate from this subspace during training. Our method, in contrast, enforces a persistent constraint, ensuring updates are strictly confined to the dynamically identified null space. This enables stable, lifelong learning within a fixed-capacity model.

## 3 METHOD: NULL SPACE ADAPTATION FOR CONTINUAL LEARNING

The core of NuSA-CL is a cyclical, data-agnostic adaptation process that enables a model to learn from a new task while preserving previously acquired knowledge. For each task in a sequence, the process unfolds in three stages as illustrated in Figure 1: **(1) Null Space Identification via SVD:** We begin with the model's current weights, $W_{t-1}$, and perform Singular Value Decomposition (SVD) to identify a low-energy subspace, that is, the intrinsic null space where prior knowledge is minimally encoded. **(2) Constrained Adaptation:** We then train a task-specific, low-rank update, $\Delta W_t$, for the current task. Crucially, this update is persistently constrained to lie strictly within the identified null space throughout training. **(3) Weight Merging:** After training, the learned update is merged directly into the backbone weights, producing the updated model $W_t \leftarrow W_{t-1} + \Delta W_t$. This evolved model,

with its fixed parameter budget, then serves as the starting point for the next task, where the cycle repeats.

## 3.1 IDENTIFYING THE INTRINSIC NULL SPACE

Let $W \in \mathbb{R}^{m \times n}$ be a weight matrix from the model. We compute its SVD, $W = U\Sigma V^\top$, to analyze its spectral properties. We posit that the principal components, associated with high-energy singular values, encode the core knowledge of the model. To provide a principled basis for our approach, we first verify the existence of a sufficiently large low-energy subspace. We identify the dimension $k$ of the principal space by finding the smallest integer that captures at least a $\rho$ fraction of the total spectral energy:

$$\sum_{i=1}^{k} \sigma_i^2 \geq \rho \cdot \|W\|_F^2 \tag{1}$$

The remaining $d - k$ dimensions constitute the intrinsic null space, spanned by the basis vectors $(U_n, V_n)$. For practical stability and to maintain a consistent number of trainable parameters across all layers and tasks, we cap the dimension of our update by a hyperparameter $r_{\max}$. The effective rank $r$ of our update is thus defined as $r = \min(d - k, r_{\max})$.

We can now express the decomposition as:

$$U = [U_p \ U_n], \quad V = [V_p \ V_n], \quad \Sigma = \begin{bmatrix} \Sigma_p & 0 \\ 0 & \Sigma_n \end{bmatrix}, \tag{2}$$

where $(U_p, V_p)$ represent the top-$k$ components of the principal subspace, and $(U_n, V_n)$ span the approximate null space.

## 3.2 CONSTRAINED ADAPTATION WITHIN THE NULL SPACE

To prevent interference with existing knowledge during continual learning, we impose a *persistent constraint* that strictly confines all new parameter updates to the identified null space. We formulate the task-specific adaptation as a LoRA-like low-rank update $\Delta W \in \mathbb{R}^{m \times n}$, but with a critical modification. Instead of learning two projection matrices, we define the update as:

$$\Delta W = U_n M V_n^\top \tag{3}$$

Here, the basis matrices $U_n$ and $V_n$ are derived from the SVD of the frozen weight $W$ and are themselves kept frozen during training for the current task. The intermediate matrix $M \in \mathbb{R}^{r \times r}$ is the only trainable component and is initialized as a zero matrix for each new task. This formulation ensures that the update $\Delta W$ is mathematically guaranteed to be orthogonal to the principal subspace of $W$, thereby minimizing interference. This persistent constraint is a key distinction from prior work (Wang et al., 2025) that uses such subspaces only for initialization, after which updates are free to deviate.

## 3.3 CONTINUAL ADAPTATION VIA UPDATE MERGING

A core component of NuSA-CL's scalability is its ability to operate within a fixed parameter budget. This is achieved by merging the learned low-rank update $\Delta W$ directly into the base weights after training on each task is complete. For a given task $t$, the new weight matrix $W_t$ is computed as:

$$W_t \leftarrow W_{t-1} + \Delta W_t \tag{4}$$

This update-and-merge cycle allows the model to sequentially accumulate knowledge from new tasks without adding any new parameters or modules. The resulting model, with its updated weights $W_t$, then becomes the starting point for the next task, $t + 1$. At the beginning of the new task, the entire process repeats: the now-updated weights $W_t$ are decomposed via SVD to identify a new intrinsic null space, ensuring that the model is always adapting in directions that are least disruptive to its full, accumulated knowledge.

## 4 THEORETICAL MOTIVATION

In this section, we provide a theoretical motivation for our approach. We analyze the degree of interference in *parameter space* to demonstrate how our persistent constraint provides a principled mechanism for mitigating catastrophic forgetting. Our analysis shows that by freezing the null space basis vectors $(U_n, V_n)$ and only learning the small intermediate matrix $M$ defined in Eq. 3, the update direction is guaranteed to be nearly orthogonal to the dominant components of the existing model weights.

### 4.1 INTERFERENCE BOUND FOR A SINGLE UPDATE

We first present a lemma that characterizes the interaction between the existing weights and a single task-specific update.

**Lemma 1** (Bounded Interference via Null Space Constraint). *Let $W = U\Sigma V^\top$ be the SVD of a weight matrix, and let $\Delta W = U_n M V_n^\top$ be an update restricted to its intrinsic null space. The interference in parameter space, measured by the Frobenius inner product, is bounded by:*

$$|\langle W, \Delta W \rangle_F| \leq \sigma_{max}^{null} \cdot \|M\|_F \tag{5}$$

*where $\sigma_{max}^{null} := \sigma_{k+1}$ is the largest singular value within the null space.*

*Proof.* Expanding the inner product: $\langle W, \Delta W \rangle_F = \mathrm{Tr}(W^\top \Delta W) = \mathrm{Tr}(V\Sigma U^\top U_n M V_n^\top)$. Since $U^\top U_n = [0; I_r]$ and $V^\top V_n = [0; I_r]$, the trace simplifies to: $\mathrm{Tr}(\Sigma_n M) \leq \|\Sigma_n\|_2 \cdot \|M\|_F = \sigma_{\max}^{null} \cdot \|M\|_F$. $\square$

### 4.2 FORGETTING CONTROL IN CONTINUAL LEARNING

The above lemma naturally generalizes to the continual learning setting, where multiple tasks are learned sequentially.

**Theorem 2** (Cumulative Interference Bound). *Let $W_t = W_{t-1} + \Delta W_t$, where $\Delta W_t = U_{t-1,n} M_t V_{t-1,n}^\top$ is the update for task $t$. The cumulative interference across $T$ tasks is bounded by:*

$$\sum_{t=1}^{T} |\langle W_{t-1}, \Delta W_t \rangle_F| \leq \sum_{t=1}^{T} \sigma_{t,max}^{null} \cdot \|M_t\|_F. \tag{6}$$

This result demonstrates that by constraining updates to low-energy subspaces, NuSA-CL bounds the cumulative parameter-level interference across tasks. This provides a principled mechanism for mitigating catastrophic forgetting, as it minimizes disruptions to the dominant weight structures that encode prior knowledge.

We emphasize that the above results are stated in parameter space and should be viewed as a local stability condition rather than a full function-level guarantee. Under standard smoothness assumptions, constraining updates in the principal directions limits how much predictions on past tasks can change, while most adaptation is pushed into the approximate null space. Empirically, Sec. 6 shows that such null-space–guided updates consistently reduce forgetting compared to alternative subspace choices and baselines. Deriving tighter, function-level forgetting bounds remains an interesting direction for future work.

## 5 EXPERIMENTS

### 5.1 EXPERIMENTAL SETUP

**Benchmarks.** Our primary evaluation is conducted on the *Multimodal Task Incremental Learning (MTIL) benchmark* (Zheng et al., 2023), a sequence of 11 diverse vision datasets designed to test a model's ability to learn new tasks while preserving its core zero-shot capabilities. To assess long-sequence scalability, we also evaluate on the standard *Class-Incremental CIFAR100* benchmark (Krizhevsky et al., 2009), splitting 100 classes into sequences of 10, 20, and 50 tasks. We report three key metrics: *Transfer*, the zero-shot accuracy on unseen tasks; *Avg.*, the average accuracy across all tasks during training; and *Last*, the final average accuracy, which measures forgetting.

Table 1: Performance and Computational Efficiency Analysis on the MTIL Benchmark. Boldface indicates the top storage-free performer. † indicates methods re-implemented on the CLIP architecture.

| Method | Computation & Memory Cost | | | | Performance (%) | | |
|---|---|---|---|---|---|---|---|
| | # Params | Additional Storage | Peak GPU (GB) | GPU-Hours | Transfer | Avg. | Last |
| *Storage-based Models* | | | | | | | |
| ZSCL (Zheng et al., 2023) | 149.6M | Data&Model (10.5GB) | 43.1 | 47.24 | 68.1 | 75.4 | 83.6 |
| MoE-Adapters (Yu et al., 2024) | 59.8M | Routers (4.8GB) | 15.5 | 3.42 | 68.9 | 76.7 | 85.0 |
| DIKI (Tang et al., 2024) | 1.8M | Task Stats (159MB) | 10.2 | 4.40 | 68.7 | 76.3 | 85.1 |
| InfLoRA† (Liang & Li, 2024) | 7.8M | Grad. Proj. Mem. (9MB) | 6.6 | 4.29 | 66.2 | 74.2 | 83.6 |
| *Storage-Free Models* | | | | | | | |
| Continual-FT | 149.6M | None | 14.6 | 12.76 | 44.6 | 55.9 | 77.3 |
| LoRA† (Hu et al., 2021) | 15.7M | None | 6.7 | **1.21** | 63.9 | 70.1 | 79.9 |
| MiLoRA† (Wang et al., 2025) | 15.7M | None | 6.7 | 1.24 | 62.8 | 68.7 | 77.4 |
| **NuSA-CL (Ours)** | **1.5M** | None | **6.6** | **1.21** | **68.6** | **75.1** | **82.8** |

Table 2: Transfer, Avg., and Last (%) for PEFT continual learning methods with a fixed backbone parameter budget on the *5-shot* MTIL benchmark. LoRA, MiLoRA, and NuSA-CL are strictly storage-free, while InfLoRA uses a small gradient projection memory; † indicates methods re-implemented on the CLIP architecture for fair comparison.

| | Aircraft | Caltech101 | CIFAR100 | DTD | EuroSAT | Flowers | Food | MNIST | OxfordPet | Cars | SUN397 | AVG. |
|---|---|---|---|---|---|---|---|---|---|---|---|---|
| Zero-shot CLIP | 24.3 | 88.4 | 68.2 | 44.6 | 54.9 | 71.0 | 88.5 | 59.4 | 89.0 | 64.7 | 65.2 | 65.3 |
| **Transfer** | | | | | | | | | | | | |
| LoRA† (Hu et al., 2021) | - | 85.9 | 63.0 | 42.3 | 39.5 | 55.9 | 80.5 | **63.7** | 76.0 | 49.5 | 57.1 | 60.4 |
| MiLoRA† (Wang et al., 2025) | - | 84.8 | 59.6 | 43.2 | 37.7 | 50.6 | 78.3 | 61.4 | 80.2 | 42.4 | 55.6 | 59.4 |
| InfLoRA† (Liang & Li, 2024) | - | 87.2 | 65.9 | **44.5** | 52.1 | 64.3 | 85.3 | 63.4 | 83.8 | 58.9 | 62.5 | 66.8 |
| **NuSA-CL (Ours)** | - | **88.2** | **67.5** | 43.3 | **55.8** | 65.3 | **86.2** | 62.8 | **84.9** | **62.2** | **64.7** | **68.1** |
| **Avg.** | | | | | | | | | | | | |
| LoRA† (Hu et al., 2021) | 15.9 | 90.6 | 68.1 | 54.1 | 69.1 | 74.3 | 81.8 | **73.6** | 81.9 | 58.3 | 62.0 | 66.8 |
| MiLoRA† (Wang et al., 2025) | 13.6 | 89.4 | 66.8 | 53.8 | 66.4 | 73.5 | 81.4 | 71.9 | 83.5 | 56.2 | 62.5 | 66.0 |
| InfLoRA† (Liang & Li, 2024) | 18.7 | **91.0** | 73.0 | 55.4 | 67.8 | **78.2** | 86.1 | 72.7 | 85.1 | 61.1 | 63.4 | 68.9 |
| **NuSA-CL (Ours)** | **28.9** | 90.5 | **73.2** | **56.1** | **71.9** | 76.9 | **87.2** | 72.9 | **86.8** | **63.5** | **65.3** | **70.3** |
| **Last** | | | | | | | | | | | | |
| LoRA† (Hu et al., 2021) | 21.3 | 89.6 | 65.3 | 58.0 | 76.8 | 83.9 | 83.2 | **90.4** | 85.6 | 67.4 | 72.1 | 72.2 |
| MiLoRA† (Wang et al., 2025) | 17.5 | 88.8 | 66.0 | 56.3 | 72.3 | 82.4 | 82.1 | 87.3 | 87.7 | 68.2 | 72.0 | 71.0 |
| InfLoRA† (Liang & Li, 2024) | 21.9 | **91.3** | 73.3 | 58.9 | 77.8 | **90.0** | 87.9 | 88.6 | 89.8 | **71.3** | **72.6** | 74.8 |
| **NuSA-CL (Ours)** | **27.2** | 90.2 | **74.0** | **59.7** | 81.3 | 85.9 | **88.9** | 90.2 | **92.0** | 69.1 | 71.2 | **75.4** |

**Implementation and Baselines.** All experiments use the CLIP ViT-B/16 backbone. Our method, NuSA-CL, identifies the null space for each task using a cumulative energy cutoff and caps the LoRA update rank at $r_{\max} = 128$. We compare NuSA-CL against three categories of baselines. (1) *Full Fine-Tuning Models* (e.g., Continual-FT, ZSCL) update all 150M parameters and require significantly more computational resources (e.g., 4 GPUs in our experiments) compared to the single GPU usage of PEFT methods. (2) *Storage-based PEFT Models* require additional, often expanding, storage, including MoE-Adapters (Yu et al., 2024), DIKI (Tang et al., 2024), and InfLoRA (Liang & Li, 2024). (3) *Storage-Free PEFT Models*, the most practical and challenging setting, includes standard LoRA (Hu et al., 2021), MiLoRA (Wang et al., 2025), and our method. For methodological relevance, we re-implemented the most comparable LoRA-based methods within a unified framework, applying adapters to both vision and text encoders with a consistent rank and merging them after each task.

## 5.2 RESULTS

**NuSA-CL Demonstrates a Superior Efficiency-Performance Tradeoff.** Table 1 presents our main results on the full-shot MTIL benchmark. As shown, storage-based PEFT methods like MoE-Adapters (Yu et al., 2024) achieve the highest final accuracy. However, this comes at a significant cost: MoE-Adapters requires nearly 60M parameters and an expanding router library, while ZSCL (Zheng

Table 3: Class-incremental CIFAR100 results: Last and Avg. accuracies (%) for 10/20/50-step splits. Bold = best, underline = second best.

| Methods | 10 steps | | 20 steps | | 50 steps | |
|---|---|---|---|---|---|---|
| | Last | Avg | Last | Avg | Last | Avg |
| CLIP (Radford et al., 2021) | 65.92 | 74.47 | 65.74 | 75.20 | 65.94 | 75.67 |
| Continual-FT | 53.23 | 65.46 | 43.13 | 59.69 | 18.89 | 39.23 |
| LwF (Li & Hoiem, 2017) | 48.04 | 65.86 | 40.56 | 60.64 | 32.90 | 47.69 |
| iCaRL (Ding et al., 2022) | 70.97 | 79.35 | 64.55 | 73.32 | 59.07 | 71.28 |
| LwF-VR (Ding et al., 2022) | 70.75 | 78.81 | 63.54 | 74.54 | 59.45 | 71.02 |
| ZSCL (Zheng et al., 2023) | 73.65 | **82.15** | 69.58 | 80.39 | 67.36 | 79.92 |
| NuSA-CL (Ours) | **74.51** | 80.25 | **73.84** | **81.03** | **71.85** | **80.19** |

et al., 2023) incurs a massive computational overhead of 47.24 GPU-Hours. In contrast, NuSA-CL establishes a new state-of-the-art within the practical and challenging storage-free setting, significantly outperforming other competitors like LoRA and MiLoRA. The key finding is that NuSA-CL achieves performance highly competitive with the storage-based SOTA while being orders of magnitude more efficient. Specifically, compared to MoE-Adapters, NuSA-CL uses 40x fewer parameters (1.5M vs. 59.8M), zero additional storage, less than half the peak GPU memory, and is nearly 3x faster (1.21 vs. 3.42 GPU-Hours). This result highlights a vastly superior performance-to-cost tradeoff, positioning NuSA-CL as a powerful and scalable solution.

**NuSA-CL Excels in Data-Efficient, Few-Shot Learning.** To further probe the robustness of our approach, we conduct a focused analysis on the challenging 5-shot MTIL benchmark, with results in Table 2. As summarized in Table 1, we distinguish two axes of practicality: (i) whether the backbone parameter budget is fixed across tasks, and (ii) whether external storage (e.g., replay buffers or gradient memories) is required. Table 2 focuses on PEFT continual learning methods that share the same CLIP backbone capacity and thus operate under a fixed backbone parameter budget. Within this regime, LoRA, MiLoRA, and NuSA-CL are strictly storage-free, whereas InfLoRA is storage-based, maintaining an additional gradient projection memory on top of the backbone. This setting acts as a stress test, magnifying the fundamental differences between each approach.

The results clearly demonstrate the superiority of our null-space adaptation strategy. NuSA-CL achieves the best performance across all summary metrics, decisively outperforming InfLoRA, the strongest competitor in this group. This indicates that our persistent null-space constraint is a fundamentally more robust and data-efficient strategy for mitigating forgetting than alternatives like subspace initialization (MiLoRA) or gradient projection (InfLoRA), validating the core mechanism of NuSA-CL.

**Scalability in Long-Sequence Incremental Learning.** Finally, to address the critical question of long-sequence scalability, we evaluate NuSA-CL on the Class-Incremental CIFAR100 benchmark. Table 3 shows that, although CLIP's zero-shot predictions already exceed naïve fine-tuning (FT) and LwF (Li & Hoiem, 2017), those baselines exhibit severe forgetting as tasks increase. Even ZSCL (Zheng et al., 2023)'s Last accuracy falls below 68% in the 50-step split. The advantage of our method becomes increasingly pronounced as the task sequence lengthens. In the most challenging 50-step scenario, NuSA-CL achieves a final 'Last' accuracy of 71.85%, significantly outperforming the strongest baseline, ZSCL, by a large margin of over 4.4%. This result provides strong empirical evidence that our dynamic, task-wise re-computation of the null space is an effective and scalable strategy for lifelong learning, confirming the longevity of our approach even after 50 sequential tasks.

## 6 ANALYSIS

In this section, we analyze the effectiveness of NuSA-CL from multiple angles. We first visualize and explain how NuSA-CL's learning dynamics—knowledge accumulation versus overwriting—fundamentally differ from conventional methods. We then establish why adapting within the null space is a superior strategy for continual learning. Finally, we experimentally validate our core mechanisms and justify the choice of key hyperparameters.

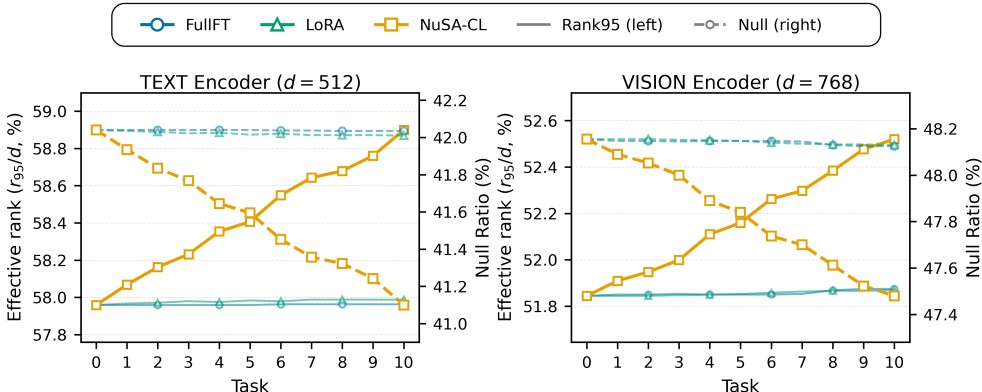

Figure 2: **Null-space dynamics across tasks.** The plots show the evolution of the effective rank (solid lines, left axis) and the null ratio (dashed lines, right axis) for the text and vision encoders. While Full-FT and LoRA exhibit nearly static spectral behavior, NuSA-CL shows a consistent increase in effective rank over tasks, indicating progressive utilization of previously underexplored spectral directions. This trend suggests that NuSA-CL integrates new knowledge by expanding into low-energy subspaces rather than overwriting dominant principal components.

### 6.1 NULL SPACE DYNAMICS: ACCUMULATION VS. OVERWRITING

Figure 2 illustrates a fundamental divergence in the learning dynamics of NuSA-CL compared to conventional fine-tuning approaches. The plots track the model's **effective rank** (solid lines), representing the capacity used to encode core knowledge, and the **null ratio** (dashed lines), the remaining underutilized capacity. The effective rank (%) is defined as the minimum percentage of dimensions required to capture 95% of the weight matrix's total spectral energy ($r_{95}/d$). $r_{95}$ is computed per layer as the smallest integer satisfying Eq. 1. The reported $r_{95}$ values in figures and tables are averaged across layers, and therefore may appear as fractional values.

Conventional methods such as LoRA and Full-FT exhibit near-static spectral behavior. As shown in Figure 2 and detailed in Appendix Table 11, their spectral properties remain almost static across all 11 tasks. For instance, the effective rank of LoRA's vision output projection layer barely changes, shifting trivially from an initial 447.42 to 447.58. This spectral inertia suggests that these methods do not exploit the model's underutilized capacity; instead, they primarily overwrite knowledge within the existing principal subspace, leaving the vast null space dormant.

In contrast, our method actively **accumulates knowledge** by progressively filling this underutilized space. For the same vision output layer, NuSA-CL's effective rank shows a clear and consistent increase. This trend, observed across all attention layers, provides direct quantitative evidence that NuSA-CL dynamically reshapes the parameter space to integrate new information. This additive learning process is the core mechanism behind NuSA-CL's ability to mitigate catastrophic forgetting and build a more informationally dense representation over time.

A natural question arises regarding the long-term viability of this approach: does the null space eventually become exhausted? Our analysis indicates that it does not. The "null space" is a low-energy spectral region, not a finite, empty container. The quantitative data in Appendix Table 11 confirms this. Even after learning 10 diverse and challenging tasks, the number of available null directions in the most saturated layer (vision output projection) is still 313.58. This is more than double our empirically chosen update rank ($r_{max} = 128$), demonstrating that a substantial, low-interference subspace persists for future adaptation.

To probe longer-horizon behavior under a highly correlated task stream, we further analyze the 50-step CIFAR-100 CIL benchmark (Table 3), where all steps are constructed from a single dataset and thus share highly correlated visual features. Our spectral analysis in Appendix Table 12 shows that the effective rank and null-space ratio measured at the beginning of Step 1 and Step 50 remain essentially stable across all attention projection matrices. These results indicate that null-space–guided updates do not induce spectral collapse even in highly correlated task streams.

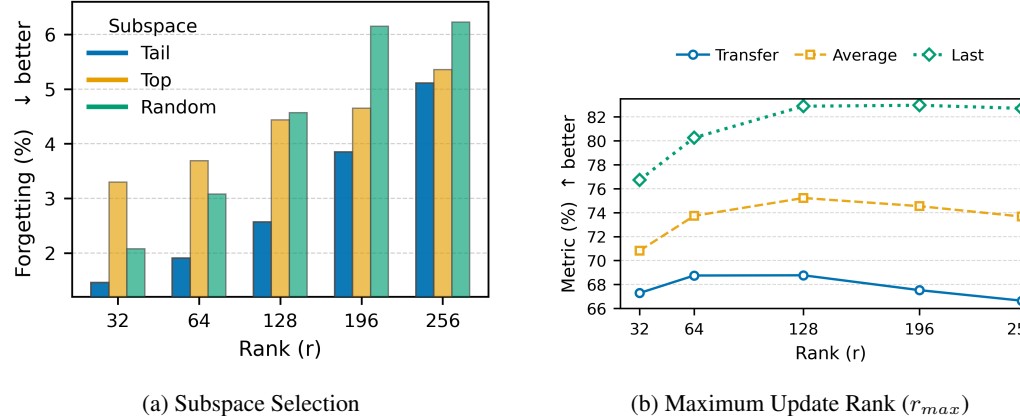

(a) Subspace Selection         (b) Maximum Update Rank ($r_{max}$)

Figure 3: **Ablations on subspace choice and update rank. (Left) Subspace selection.** Across all ranks, the *Tail* (null-like) subspace consistently yields lower forgetting than *Top* and *Random*, indicating that low-energy directions provide a low-interference region for continual updates. **(Right) Maximum update rank.** Within the *Tail* subspace, overall continual learning performance is maximized at $r_{\max} = 128$ (measured by Transfer, Avg., and Last), suggesting that a moderate rank balances retention (stability) and task adaptation (plasticity).

## 6.2 WHY THE NULL SPACE? SUBSPACE SELECTION STRATEGY

**Tail outperforms Top and Random in mitigating forgetting.** To identify an update subspace that minimizes catastrophic forgetting, we analyze how different spectral directions affect knowledge retention and adaptation. We compare three low-rank initialization strategies—*Top* (largest singular directions), *Tail* (smallest, null-like directions), and *Random*—using a fixed rank $r = 128$ and 1000 training iterations per task on the 11 MTIL datasets. Forgetting (defined as the average drop from post-task to final performance) is reported in Fig. 3a, with detailed per-rank results in Appx. 10.

Across all tested ranks, the *Tail* strategy consistently yields the lowest forgetting. For example, at $r = 128$, *Tail* achieves 2.57% forgetting compared to 4.44% for *Top* and 4.57% for *Random*. Although forgetting gradually increases with larger ranks (from 1.46% at $r = 32$ to 5.11% at $r = 256$), *Tail* remains superior at every rank. These results indicate that low-energy directions form a safer region for continual updates, reducing interference with previously acquired knowledge.

**Balancing stability and plasticity for continual learning.** Does this imply that "the smaller (purer) the null space, the better" for continual learning? Not necessarily. As shown in Fig. 3b, overall CL performance is maximized around an update rank of $r_{\max}=128$, which balances stability (retention) and plasticity (on-task adaptation). Consistent with Table 10, we observe a stability–plasticity trade-off at *both* the update-rank dimension and the subspace choice: increasing rank improves **Target** performance (accuracy on the currently learned task) but induces more forgetting, and *Top* attains marginally higher **Target** accuracy on the current task yet suffers substantially larger forgetting. Because continual learning is ultimately constrained by retention, these results motivate our design in NuSA-CL: operate in the null space (*Tail*) and cap the update rank at $r_{\max}$ to maintain stability while preserving competitive adaptation.

## 6.3 VALIDATION OF NUSA-CL'S DESIGN PRINCIPLES

**Core Mechanisms are Critical.** As shown in Table 4a, our core design choices are essential for success. The *persistent constraint* is vital; unfreezing the null space bases ($U_n, V_n$) to make them trainable leads to a significant drop in performance, confirming that a strict, persistent constraint is necessary to prevent forgetting. Similarly, *multimodal adaptation* is superior, as jointly updating both text and vision encoders is key to maintaining cross-modal alignment.

**Practicality and Robustness.** A potential concern for our method is the overhead of SVD and sensitivity to hyperparameters. However, our analysis shows NuSA-CL is both practical and robust,

Table 4: Ablation studies validating NuSA-CL's design principles. (a) Core mechanisms such as the persistent constraint and multimodal adaptation are shown to be critical. (b) The method is practical, with negligible initialization overhead, and robust to hyperparameter choices.

(a) Core Mechanism Ablations

**Persistent Constraint Ablation**

| Condition | Transfer | Avg. | Last |
|---|---|---|---|
| Train only M (Ours) | **68.58** | **75.08** | **82.79** |
| Train M & $V_n$ | 66.37 | 73.11 | 82.04 |
| Train M, $U_n$, $V_n$ | 62.60 | 68.12 | 77.32 |

**Modality Ablation**

| Modality | Transfer | Avg. | Last |
|---|---|---|---|
| Both (Ours) | **68.58** | **75.08** | **82.79** |
| Text-only | 68.47 | 72.62 | 79.09 |
| Vision-only | 65.14 | 70.49 | 77.86 |

(b) Robustness and Practicality

**Robustness to Cutoff Threshold ($\rho$)**

| threshold | Transfer | Avg. | Last |
|---|---|---|---|
| 0.80 | 68.29 | 74.87 | 82.28 |
| 0.90 | **68.82** | 75.07 | 82.74 |
| **0.95** | 68.58 | **75.08** | **82.79** |
| 0.99 | 68.49 | 74.89 | 82.70 |
| 0.999 | 68.11 | 72.89 | 79.16 |

**SVD Efficiency Analysis**

| Method | Init. Time | Train Time (hr) | Avg. |
|---|---|---|---|
| InfLoRA | $\sim$81 min | 4.29 | 74.2 |
| **Ours** | **<1 min** | **1.21** | **75.1** |

shown in Table 4b. The SVD initialization is exceptionally lightweight. While our data-agnostic SVD is a one-time calculation per task with negligible overhead, competing methods like InfLoRA require heavy, data-dependent computations that scale poorly to design subspace using training dataset before learning. Furthermore, results show that performance is remarkably stable across a wide range of energy cutoff thresholds, demonstrating that NuSA-CL does not require sensitive hyperparameter tuning.

**Practical guidance for larger backbones.** In practice, we compute SVD once per task and per layer on the attention projection matrices (e.g., $W_q, W_k, W_v, W_o$), whose sizes remain moderate for CLIP-style backbones (e.g., up to $512 \times 512$ for text and $768 \times 768$ for vision in our ViT-B/16 experiments), making the spectral step inexpensive compared to standard forward–backward passes. While our empirical results focus on ViT-B/16, the same scaling behavior extends naturally to larger CLIP variants where attention projections typically have dimensionalities on the order of 768–1024. In such cases, practitioners may adopt truncated or approximate SVD on selected layers to further reduce overhead; these engineering choices are orthogonal to the core method and primarily affect runtime rather than the underlying learning dynamics.

## 7  CONCLUSION

This paper tackles the challenge of adapting vision-language models to evolving tasks without catastrophic forgetting and without the unsustainable resource costs of methods whose storage or parameter counts grow linearly with the number of tasks, rendering them impractical for lifelong learning. We introduce **NuSA-CL**, a memory-free framework based on intrinsic adaptation. NuSA-CL identifies underutilized null space directions and constrains low-rank updates to this subspace, integrating new knowledge while preserving pre-trained capabilities. The learned update is then merged into the base model, maintaining a fixed parameter budget. Across benchmarks, our method delivers a superior performance-efficiency trade-off: it outperforms other storage-free methods and rivals resource-intensive, storage-based approaches at a fraction of the cost. Strong results on long task sequences validate its scalability and effectiveness for lifelong learning, positioning NuSA-CL as a practical solution for deploying adaptable vision–language models in resource-constrained settings.

**Limitations and future work.** NuSA-CL remains robust on sequences of up to 50 tasks with ViT-B, but its capacity under extreme lifelong settings where the available null space directions may saturate warrants further study. We also note that the SVD step, while negligible with our reduced SVD on ViT-B, could become a bottleneck for substantially larger models. Future work includes (i) quantifying sensitivity to task order and semantic relatedness, as highly correlated sequences may concentrate usage of specific null-space dimensions; and (ii) developing lightweight, more reversible integration strategies, enabling selective forgetting without relying on persistent external memory.

**Acknowledgements.** This work was supported by Institute of Information & communications Technology Planning & Evaluation(IITP) under the Leading Generative AI Human Resources Development(IITP-2026-RS-2024-00397085) grant funded by the Korea government(MSIT). This work was also supported by the National Research Foundation of Korea (NRF) grant funded by the Korea government (MSIT) (No. RS-2024-00345809, Research on AI Robustness Against Distribution Shift in Real-World Scenarios; and No. RS-2023-00222663, Center for Optimizing Hyperscale AI Models and Platforms).

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

# APPENDIX

## A  DATASET & IMPLEMENTATION DETAILS

### A.1  BENCHMARKS AND METRICS.

Our primary evaluation is conducted on the *Multimodal Task Incremental Learning (MTIL) benchmark* (Zheng et al., 2023), which requires a model to sequentially train on 11 tasks while maintaining CLIP's zero-shot performance. The benchmark comprises Aircraft (Maji et al., 2013), Caltech101 (Fei-Fei et al., 2004), CIFAR100 (Krizhevsky et al., 2009), DTD (Cimpoi et al., 2014), EuroSAT (Helber et al., 2019), Flowers (Nilsback & Zisserman, 2008), Food (Bossard et al., 2014), MNIST (Deng, 2012), OxfordPet (Parkhi et al., 2012), StanfordCars (Krause et al., 2013), and SUN397 (Xiao et al., 2010). Appendix Table 5 summarizes the number of classes, training/test images, and recognition type for each task, highlighting the scale and heterogeneity of MTIL. We report three metrics after completing the entire sequence:

- **Transfer:** Measures zero-shot transfer capability. After training on task $t$, we evaluate on the test sets of all future, unseen tasks $t + 1, \ldots, 11$ and average the results.
- **Avg.:** The mean test accuracy across all 11 datasets, recorded immediately after training on each task.
- **Last:** The mean test accuracy of the final model on each task's test set, capturing performance degradation (forgetting).

Table 5: Dataset statistics for the MTIL benchmark (Zheng et al., 2023). The 11 tasks span diverse recognition problems and domains, from fine-grained aircraft and car series to satellite imagery and scene categorization.

| Dataset | # classes | # train | # test | Recognition task |
|---|---|---|---|---|
| Aircraft (Maji et al., 2013) | 100 | 3334 | 3333 | aircraft series |
| Caltech101 (Fei-Fei et al., 2004) | 101 | 6941 | 1736 | real-world objects |
| CIFAR100 (Krizhevsky et al., 2009) | 100 | 50000 | 10000 | real-world objects |
| DTD (Cimpoi et al., 2014) | 47 | 1880 | 1880 | texture recognition |
| EuroSAT (Helber et al., 2019) | 10 | 21600 | 5300 | satellite scenes |
| Flowers (Nilsback & Zisserman, 2008) | 102 | 1020 | 6149 | flower species |
| Food (Bossard et al., 2014) | 101 | 75750 | 25250 | food types |
| MNIST (Deng, 2012) | 10 | 60000 | 10000 | digit recognition |
| OxfordPet (Parkhi et al., 2012) | 37 | 3680 | 3669 | animal species |
| StanfordCars (Krause et al., 2013) | 196 | 8144 | 8041 | car series |
| SUN397 (Xiao et al., 2010) | 397 | 87003 | 21751 | scene categories |
| Total | 1201 | 319352 | 97109 | – |

To test for long-sequence scalability, we also evaluate on the standard CIFAR100 class-incremental splits (Krizhevsky et al., 2009). To ensure comparability with prior work, we adopt the same protocol and report the baseline numbers in ZSCL (Zheng et al., 2023). The 100 classes are grouped into 10, 20, or 50 tasks (10, 5, or 2 classes per task respectively). As the number of tasks increases, catastrophic forgetting becomes more severe. Complementary comparisons to CLIP-based PEFT continual-tuning methods such as DIKI and MoE-Adapters are provided on the MTIL benchmark, which is their primary evaluation setting, while CIFAR100 serves here as a long-horizon CIL stress test.

### A.2  IMPLEMENTATION DETAILS.

All experiments use the CLIP ViT-B/16 backbone with adapters applied to every query, key, value, and output projection. For the methods re-implemented on the CLIP architecture, we follow the CLIP-LoRA implementation (Zanella & Ayed, 2024). For a rigorous and fair comparison across all LoRA like methods (LoRA, MiLoRA, InfLoRA, and NuSA-CL), we implemented them within an identical framework: adapters were applied to both vision and text encoders with a consistent rank, and the updated weights were merged into the backbone after each task.

Table 6: Accuracy (%) of our method on the MTIL benchmark (5-shot, 500 iterations). Metrics for the Transfer, Last and Avg. are shown in the rightmost column.

| | Aircraft | Caltech101 | CIFAR100 | DTD | EuroSAT | Flowers | Food | MNIST | OxfordPet | StanfordCars | SUN397 | Metric |
|---|---|---|---|---|---|---|---|---|---|---|---|---|
| Transfer | | 88.2 | 67.5 | 43.3 | 55.8 | 65.3 | 86.2 | 62.8 | 84.9 | 62.2 | 64.7 | **68.1** |
| Aircraft | 33.4 | 88.2 | 68.2 | 44.4 | 56.2 | 66.1 | 87.4 | 57.3 | 85.9 | 63.9 | 65.8 | |
| Caltech101 | 28.6 | 91.8 | 66.8 | 41.9 | 53.4 | 53.1 | 82.9 | 57.3 | 72.6 | 62.1 | 64.0 | |
| CIFAR100 | 31.1 | 91.4 | 76.1 | 43.7 | 57.3 | 70.2 | 86.9 | 62.5 | 86.1 | 62.1 | 66.7 | |
| DTD | 30.0 | 90.9 | 75.6 | 62.1 | 56.2 | 68.7 | 87.0 | 66.4 | 87.5 | 62.5 | 66.0 | |
| EuroSAT | 29.2 | 90.8 | 75.1 | 62.0 | 82.4 | 68.2 | 86.3 | 66.2 | 86.6 | 62.2 | 65.2 | |
| Flowers | 28.2 | 91.1 | 74.4 | 61.5 | 80.9 | 88.1 | 86.6 | 65.2 | 86.7 | 62.0 | 64.9 | |
| Food | 28.1 | 90.8 | 74.0 | 60.9 | 81.6 | 86.7 | 88.6 | 64.8 | 86.6 | 61.9 | 64.3 | |
| MNIST | 27.7 | 90.6 | 73.9 | 60.8 | 81.0 | 87.2 | 88.5 | 90.7 | 87.1 | 61.9 | 64.4 | |
| OxfordPet | 27.3 | 89.6 | 73.5 | 60.1 | 80.1 | 85.5 | 88.2 | 90.6 | 91.8 | 61.2 | 61.9 | |
| StanfordCars | 27.0 | 89.8 | 73.6 | 60.3 | 80.0 | 86.5 | 88.3 | 90.6 | 91.7 | 69.1 | 64.1 | |
| SUN397 | 27.2 | 90.2 | 74.0 | 59.7 | 81.3 | 85.9 | 88.9 | 90.2 | 92.0 | 69.1 | 71.2 | **75.4** |
| Average | 28.9 | 90.5 | 73.2 | 56.1 | 71.9 | 76.9 | 87.2 | 72.9 | 86.8 | 63.5 | 65.3 | **70.3** |

Table 7: Accuracy (%) of our method on the MTIL benchmark (full dataset, 1000 iterations). The rightmost column shows the overall Transfer, Last, and Average metrics.

| | Aircraft | Caltech101 | CIFAR100 | DTD | EuroSAT | Flowers | Food | MNIST | OxfordPet | StanfordCars | SUN397 | Metric |
|---|---|---|---|---|---|---|---|---|---|---|---|---|
| Transfer | | 88.3 | 66.8 | 44.0 | 55.5 | 67.9 | 85.8 | 66.7 | 84.8 | 60.7 | 65.2 | **68.6** |
| Aircraft | 49.7 | 88.3 | 67.5 | 44.3 | 53.6 | 69.9 | 88.3 | 58.0 | 87.7 | 63.2 | 65.5 | |
| Caltech101 | 42.9 | 96.7 | 66.0 | 42.3 | 53.2 | 64.5 | 85.2 | 55.5 | 83.9 | 60.7 | 63.8 | |
| CIFAR100 | 41.0 | 96.1 | 82.2 | 45.4 | 59.8 | 69.2 | 86.2 | 73.7 | 85.1 | 61.6 | 66.4 | |
| DTD | 41.6 | 96.2 | 81.6 | 74.2 | 55.6 | 68.2 | 85.6 | 71.2 | 84.6 | 61.5 | 65.8 | |
| EuroSAT | 39.9 | 95.6 | 81.0 | 73.6 | 97.0 | 67.9 | 85.0 | 69.7 | 84.3 | 60.6 | 65.1 | |
| Flowers | 38.8 | 95.7 | 80.6 | 72.1 | 96.9 | 96.4 | 84.8 | 69.9 | 82.9 | 60.1 | 64.7 | |
| Food | 38.6 | 95.5 | 80.7 | 73.5 | 96.9 | 95.9 | 91.1 | 68.7 | 85.4 | 60.3 | 65.7 | |
| MNIST | 34.3 | 95.9 | 79.8 | 72.4 | 96.6 | 95.9 | 91.1 | 98.9 | 84.9 | 59.6 | 65.7 | |
| OxfordPet | 33.2 | 95.9 | 79.6 | 72.3 | 96.6 | 95.2 | 90.6 | 98.9 | 94.8 | 59.0 | 64.4 | |
| StanfordCars | 33.2 | 95.6 | 79.7 | 71.8 | 96.6 | 94.6 | 90.6 | 98.9 | 95.1 | 79.2 | 64.9 | |
| SUN397 | 35.0 | 95.2 | 79.4 | 71.5 | 96.2 | 94.0 | 90.6 | 98.8 | 95.0 | 78.1 | 76.9 | **82.8** |
| Average | 38.9 | 95.2 | 78.0 | 64.9 | 81.7 | 82.9 | 88.1 | 78.4 | 87.6 | 64.0 | 66.3 | **75.1** |

We use the AdamW optimizer (learning rate $3 \times 10^{-4}$, weight decay $10^{-2}$, $\beta_1 = 0.9$, $\beta_2 = 0.999$), with a linear learning rate warmup for the first 5% of iterations followed by a cosine decay schedule. For NuSA-CL, we identify the null-space using a cumulative energy ratio cutoff (99% for 5-shot, 95% for full-shot) and cap the maximum rank at $r_{\max} = 128$.

In the *5-shot MTIL setting* (Yu et al., 2024), we sample five examples per class and train for 500 iterations with a batch size of 16. We scale each low-rank update by $\alpha/\sqrt{r}$ with $\alpha = 1$, apply a dropout rate of 0.25 to the adapter branches, and use label smoothing of 0.2. In the *full-shot setting*, we use all training samples, train for 1000 iterations, and use a scaling factor of $\alpha = 2$, keeping all other hyperparameters identical. All experiments were conducted on a single NVIDIA RTX 3090 GPU. In the CIL setting, we use a learning rate of $3 \times 10^{-3}$, maximum rank $r_{\max} = 256$, dropout of 0.05, batch size 128, and no label smoothing.

## A.3 BASELINES.

We evaluate NuSA-CL against baselines along two complementary axes: (i) whether additional storage is required, and (ii) whether the backbone parameter budget is fixed across tasks.

Under the storage criterion (Table 1), methods are divided into storage-based and storage-free approaches. Storage-based methods include *ZSCL* (Zheng et al., 2023), which uses knowledge distillation, *MoE-Adapters* (Yu et al., 2024), *DIKI* (Tang et al., 2024), and *InfLoRA* (Liang & Li, 2024). Although originally proposed for ViTs, we adapt InfLoRA's gradient projection memory

Table 8: Transfer, Avg., and Last (%) for continual learning methods on the Order-2 sequence of the *5-shot* MTIL benchmark. † indicates methods re-implemented on the CLIP architecture.

| | Cars | Food | MNIST | OxfordPet | Flowers | SUN397 | Aircraft | Caltech101 | DTD | EuroSAT | CIFAR100 | AVG. |
|---|---|---|---|---|---|---|---|---|---|---|---|---|
| Zero-shot | 64.7 | 88.5 | 59.4 | 89.0 | 71.0 | 65.2 | 24.3 | 88.4 | 44.6 | 54.9 | 68.2 | 65.3 |
| **Transfer** | | | | | | | | | | | | |
| LoRA† (Hu et al., 2021) | | 87.6 | **63.0** | 86.6 | 63.5 | 63.2 | 19.8 | 87.4 | 43.8 | 44.0 | 61.3 | 62.0 |
| MiLoRA† (Wang et al., 2025) | | 88.1 | 62.9 | **87.4** | **87.4** | 62.6 | 18.3 | 86.8 | 41.0 | 45.1 | 59.2 | 61.4 |
| InfLoRA† (Liang & Li, 2024) | | **88.2** | 58.8 | 84.1 | 65.4 | 65.4 | 20.7 | 87.7 | **44.2** | 49.3 | **66.9** | 62.9 |
| **NuSA-CL (Ours)** | | 87.6 | 60.0 | 86.3 | 65.8 | 63.8 | 21.9 | 88.3 | 43.6 | 53.8 | 68.3 | 63.9 |
| **Avg.** | | | | | | | | | | | | |
| LoRA† (Hu et al., 2021) | 55.8 | 80.3 | **86.5** | 84.3 | 72.5 | 66.3 | 21.9 | 88.4 | **48.3** | 50.1 | 62.4 | 65.2 |
| MiLoRA† (Wang et al., 2025) | 51.0 | 76.7 | 83.8 | 81.7 | 71.7 | 64.3 | 19.2 | 87.7 | 44.1 | 50.7 | 60.1 | 62.8 |
| InfLoRA† (Liang & Li, 2024) | 65.3 | **85.5** | 85.3 | 85.6 | **80.4** | 67.1 | 25.3 | 89.3 | 48.3 | 54.4 | 67.6 | 68.6 |
| **NuSA-CL (Ours)** | **66.3** | 87.6 | 84.1 | 89.9 | 78.5 | **67.3** | 27.1 | 89.4 | 47.5 | **58.5** | 68.9 | 69.6 |
| **Last** | | | | | | | | | | | | |
| LoRA† (Hu et al., 2021) | 46.7 | 76.7 | 89.3 | 82.1 | 71.3 | 67.9 | 23.7 | 90.0 | 59.1 | 72.4 | 73.3 | 68.4 |
| MiLoRA† (Wang et al., 2025) | 28.6 | 63.1 | 69.5 | 73.0 | 58.2 | 61.6 | 14.0 | 87.7 | 49.1 | 71.6 | 69.0 | 58.7 |
| InfLoRA† (Liang & Li, 2024) | 60.4 | 83.3 | **90.0** | 86.6 | **87.1** | 69.6 | 30.7 | 91.2 | **59.4** | 76.7 | 73.7 | 73.5 |
| **NuSA-CL (Ours)** | **64.6** | **86.7** | 89.4 | **91.7** | 84.7 | **70.1** | **32.7** | **91.8** | 58.0 | **79.2** | **75.1** | **74.9** |

Table 9: Transfer, Avg., and Last (%) for continual learning methods on the Order-2 sequence of the the full dataset MTIL benchmark. † indicates methods re-implemented on the CLIP architecture.

| | Cars | Food | MNIST | OxfordPet | Flowers | SUN397 | Aircraft | Caltech101 | DTD | EuroSAT | CIFAR100 | AVG. |
|---|---|---|---|---|---|---|---|---|---|---|---|---|
| Zero-shot | 64.7 | 88.5 | 59.4 | 89.0 | 71.0 | 65.2 | 24.3 | 88.4 | 44.6 | 54.9 | 68.2 | 65.3 |
| **Transfer** | | | | | | | | | | | | |
| LoRA† (Hu et al., 2021) | | 88.2 | 59.7 | 86.9 | 65.9 | 65.1 | 20.7 | 88.3 | 44.8 | 58.6 | 62.0 | 62.0 |
| MiLoRA† (Wang et al., 2025) | | 88.2 | 61.9 | **87.7** | 63.3 | 64.5 | 19.1 | 87.6 | 47.0 | 63.4 | 62.6 | 62.6 |
| InfLoRA† (Liang & Li, 2024) | | **88.2** | **62.1** | 86.7 | 65.2 | **66.0** | 20.9 | 88.3 | **45.1** | 65.2 | 63.4 | 63.4 |
| **NuSA-CL (Ours)** | | 88.1 | 57.7 | 87.0 | 66.2 | 64.8 | 21.9 | 89.2 | 49.4 | 66.9 | 63.4 | 63.4 |
| **Avg.** | | | | | | | | | | | | |
| LoRA† (Hu et al., 2021) | 71.6 | 85.2 | 91.8 | 91.4 | 77.9 | 70.2 | 18.6 | 91.5 | 53.3 | 51.0 | 60.9 | 69.4 |
| MiLoRA† (Wang et al., 2025) | 67.0 | 85.3 | 92.3 | 90.9 | 76.0 | 69.6 | 20.2 | 90.8 | 51.8 | 55.9 | 65.3 | 69.6 |
| InfLoRA† (Liang & Li, 2024) | 80.0 | **89.2** | **92.4** | **92.2** | 82.8 | 71.9 | 30.8 | 91.5 | 53.5 | 55.4 | 67.0 | **73.3** |
| **NuSA-CL (Ours)** | 73.8 | 89.7 | 91.2 | 92.1 | **84.4** | 70.8 | 32.9 | 91.6 | 51.3 | 58.0 | 68.2 | 73.1 |
| **Last** | | | | | | | | | | | | |
| LoRA† (Hu et al., 2021) | 52.5 | 78.1 | 96.8 | 91.9 | 72.5 | 71.6 | 3.9 | **97.0** | 73.4 | 84.7 | 84.3 | 73.3 |
| MiLoRA† (Wang et al., 2025) | 42.5 | 77.1 | 97.7 | 87.5 | 65.6 | 69.6 | 13.2 | 96.1 | 72.5 | 93.2 | 84.2 | 72.6 |
| InfLoRA† (Liang & Li, 2024) | **75.1** | 87.2 | **98.4** | 93.6 | 88.1 | 75.5 | 30.7 | 96.4 | **74.8** | 95.5 | **84.7** | 82.2 |
| **NuSA-CL (Ours)** | 64.6 | **88.6** | **98.4** | 93.4 | **93.9** | 75.8 | 44.8 | 95.8 | 72.5 | **96.4** | 80.9 | **82.9** |

mechanism to the multimodal CLIP architecture for conceptual alignment. Storage-free methods include *Continual-FT*, which naively fine-tunes all parameters, the standard *LoRA* (Hu et al., 2021), *MiLoRA* (Wang et al., 2025), an LLM adaptation method we implement on CLIP due to the relevance of its subspace initialization strategy, and our proposed *NuSA-CL*.

Under the fixed-backbone PEFT setting (Table 2), we compare LoRA-like methods implemented within an identical framework to ensure a controlled and fair comparison.

Table 10: Per-rank results by subspace type. Lower is better for *Forgetting*; higher is better for *Target*.

| Rank | Subspace | Forgetting (%) | Target (%) |
|------|----------|----------------|------------|
|      | Tail     | 1.46           | 78.05      |
| 32   | Top      | 3.30           | 79.66      |
|      | Random   | 2.08           | 78.70      |
|      | Tail     | 1.91           | 81.99      |
| 64   | Top      | 3.69           | 82.96      |
|      | Random   | 3.08           | 82.77      |
|      | Tail     | 2.57           | 85.22      |
| 128  | Top      | 4.44           | 85.50      |
|      | Random   | 4.57           | 85.49      |
|      | Tail     | 3.85           | 86.46      |
| 196  | Top      | 4.65           | 86.76      |
|      | Random   | 6.15           | 86.67      |
|      | Tail     | 5.11           | 87.34      |
| 256  | Top      | 5.36           | 87.46      |
|      | Random   | 6.23           | 87.44      |

## B  ADDITIONAL EVALUATION RESULTS

### B.1  MTIL COMPLETE RESULTS

Table 6 reports the results on all eleven tasks in the 5-shot MTIL regime (500 iterations per task). Our method maintains strong zero-shot retention (Transfer = 68.1%), while achieving an Average accuracy of 70.3% and a Last accuracy of 75.4%. The per-dataset breakdown confirms that our method uniformly preserves performance: no task suffers a dramatic collapse, and gains over baselines appear across both domain-shifted benchmarks (e.g., EuroSAT, Flowers) and in-domain benchmarks (e.g., CIFAR100, MNIST). Table 7 presents the analogous results in the full-shot regime (1000 iterations per task), showing the same pattern of robust Transfer, Avg., and Last scores.

### B.2  MTIL ORDER-2 RESULTS

To assess sensitivity to task ordering, Table 8 and Table 9 report 5-shot and full-shot MTIL results, respectively, under a different task sequence (Order-2). In both settings, our method again achieves the highest Transfer, Average, and Last metrics, matching the original ordering (Order-1). Crucially, Last accuracy remains above 80% even in this permuted protocol, confirming that our method mitigates forgetting regardless of task order. This order-agnostic stability underscores the general applicability of our approach.

## C  FURTHER ABLATION

This section provides detailed results for the analysis in Section 6. Table 10 presents the numerical data corresponding to the observations in Figure 3a. Table 11 lists the specific dimension numbers plotted in Figure 2. Table 12 further reports the effective ranks and null-space fractions of NuSA-CL on the 50-step CIFAR-100 CIL benchmark, illustrating that the spectra remain stable even after long-horizon continual updates.

## D  LLM USAGE STATEMENT

We disclose that Large Language Models (LLMs) were utilized as an auxiliary tool in the preparation of this manuscript. The use of LLMs was limited to writing assistance, such as improving the clarity of English expressions, correcting grammatical errors, and refining sentence structure for better readability. Specifically, LLMs were employed to rephrase complex sentences for conciseness and to

Table 11: Effective rank $r_{95}$ and corresponding null directions (Null@95 = $d - r_{95}$) for CLIP, LoRA and NuSA-CL across encoders and projection parameters. For CLIP, we report spectra before any MTIL task is learned. For LoRA and NuSA-CL, we report spectra measured at the beginning of the final MTIL task (Task 11), after sequentially training on all preceding tasks.

| Method | Encoder | Param | $r_{95}$ | Null@95 |
|---|---|---|---|---|
| CLIP | TEXT | q | 279.67 | 232.33 |
| | | k | 284.83 | 227.17 |
| | | v | 311.33 | 200.67 |
| | | o | 311.17 | 200.83 |
| | VISION | q | 354.08 | 413.92 |
| | | k | 358.42 | 409.58 |
| | | v | 432.75 | 335.25 |
| | | o | 447.42 | 320.58 |
| LoRA (Task 11) | TEXT | q | 279.75 | 232.25 |
| | | k | 284.92 | 227.08 |
| | | v | 311.58 | 200.42 |
| | | o | 311.33 | 200.67 |
| | VISION | q | 354.17 | 413.83 |
| | | k | 358.75 | 409.25 |
| | | v | 432.92 | 335.08 |
| | | o | 447.58 | 320.42 |
| NuSA-CL (Task 11) | TEXT | q | 282.58 | 229.42 |
| | | k | 288.25 | 223.75 |
| | | v | 316.67 | 195.33 |
| | | o | 318.75 | 193.25 |
| | VISION | q | 357.75 | 410.25 |
| | | k | 362.42 | 405.58 |
| | | v | 438.83 | 329.17 |
| | | o | 454.42 | 313.58 |

Table 12: Spectral stability of NuSA-CL on CIFAR-100 CIL (50 steps). We report the effective rank $r_{95}$ and the fraction of the approximate null space (*Null ratio* = $(d - r_{95})/d$) measured at the beginning of Step 1 (before any continual updates) and at the beginning of Step 50 (after learning the first 49 steps).

| Encoder | Param | $r_{95}$ (Step 1) | $r_{95}$ (Step 50) | Null ratio (Step 1) | Null ratio (Step 50) |
|---|---|---|---|---|---|
| TEXT | q | 279.67 | 279.67 | 45.38% | 45.38% |
| | k | 284.83 | 285.00 | 44.37% | 44.34% |
| | v | 311.33 | 311.83 | 39.19% | 39.10% |
| | o | 311.17 | 311.67 | 39.23% | 39.13% |
| VISION | q | 354.08 | 354.17 | 53.90% | 53.88% |
| | k | 358.42 | 358.50 | 53.33% | 53.32% |
| | v | 432.75 | 432.75 | 43.65% | 43.65% |
| | o | 447.42 | 447.67 | 41.74% | 41.71% |

receive suggestions for appropriate vocabulary fitting an academic tone. All scientific contributions including the core research ideas, experimental design, analysis of results, and drawing of conclusions were made entirely by the authors. The LLMs did not influence the originality or the core scientific content of this work.

