# OpenReview forum: "Memory-Free Continual Learning with Null Space Adaptation for Zero-Shot Vision-Language Models"
_ICLR.cc/2026/Conference — ICLR 2026 Poster_

### Official Review · Reviewer_TeKM · 2025-10-17

**Soundness:** 2
**Presentation:** 3
**Contribution:** 2
**Rating:** 4
**Confidence:** 3

**Summary:**

This paper proposes NuSA-CL, a memory-free continual learning method that leverages orthogonal weight modulation to mitigate catastrophic forgetting. The approach updates model parameters only along directions orthogonal to previously learned subspaces, thus avoiding interference with past knowledge without storing any exemplars. Experiments on several vision and vision-language benchmarks show that NuSA-CL performs competitively among storage-free approaches but lags behind replay-based methods.

**Strengths:**

1.	Memory-free design.
The method achieves continual learning without any replay or exemplar storage, offering a clean and theoretically motivated direction for efficient CL.
2.	Strong performance among storage-free models.
Within the memory-free setting, NuSA-CL demonstrates solid results and stable learning behavior across multiple benchmarks.

**Weaknesses:**

1.	Limited performance.
The method performs worse than all storage-based models, showing limited competitiveness in practical CL scenarios.
2.	Simplicity vs. effectiveness.
The proposed approach is simple, but its results do not convincingly show that such a minimal mechanism can achieve strong continual learning.
3.	Scalability concern.
As also noted by the authors, under extreme lifelong settings the null-space directions may saturate. Without a clear advantage in scalability over storage-based methods, the claimed benefit of being memory-free remains modest.

**Questions:**

1.	What prevents NuSA-CL from surpassing any storage-based methods in performance, despite its theoretically clean orthogonal design?

---

> ### Author Response · Authors · 2025-11-21
> **Response to Reviewer TeKM (1/2)**
>
> We thank Reviewer TeKM for the thoughtful review and for recognizing the value of our memory-free design. Regarding the performance gap to storage-based methods, we view this as a fundamental trade-off between **capacity expansion** and **fixed-budget efficiency**, rather than a limitation of the method itself. Below we address your questions point by point.
>
> ### **1. Performance relative to storage-based methods (W1 & Q1)**
> We first clarify that storage-free NuSA-CL remains competitive with, and in some cases surpasses, storage-based methods even under a strictly memory-free regime and a fixed backbone parameter budget.
>
> In the 5-shot MTIL setting (Table 2), NuSA-CL achieves higher Transfer/Avg./Last accuracies (68.1 / 70.3 / 75.4) than InfLoRA (66.8 / 68.9 / 74.8), despite InfLoRA relying on a gradient projection memory. This shows that, under a PEFT continual-learning setting with a fixed backbone parameter budget (all methods use the same backbone), null-space adaptation alone can match or exceed the benefits of explicit gradient replay. On the CIFAR-100 CIL benchmark (Table 4), NuSA-CL also attains higher Last accuracy than ZSCL, even though ZSCL is a full fine-tuning baseline that uses knowledge distillation with reference data, indicating robustness under very long task sequences.
>
> We agree that there is a trade-off when one is allowed to maintain task-wise modules or rehearsal buffers in continual learning. When storage-based methods win in the full-shot MTIL setting, this is largely because they allocate task-wise capacity to isolate knowledge via task-specific adapters, routers, or other parameters, which generally require additional memory that grows with the number of tasks. In contrast, NuSA-CL aims to make more effective use of the existing model capacity, instead of relying on extra modules or buffers. It maximizes useful expressivity under a fixed parameter budget by activating under-utilized capacity while preserving previously important directions.
>
> Beyond the raw memory footprint, storage-based methods that rely on task-wise modules such as MoE-Adapters and DIKI also depend on **explicit task identification** at inference time (e.g., DDAS in MoE, distribution matching in DIKI). This introduces a discrete **routing decision**: the system must first answer “which task is this?” before choosing the appropriate expert or prompt. Such routing typically requires checks against all $T$ tasks and can be brittle—if the router misclassifies an input (especially in ambiguous or unseen domains), the wrong expert is activated, which may substantially degrade performance. By contrast, NuSA-CL requires **no task identification** at inference. All inputs are processed by a single, continually updated backbone, which is advantageous in real-world, open-ended scenarios where task boundaries are blurred or not known a priori.
>
> ---
>
> ### **2. On scalability and saturation in extreme lifelong settings (W3)**
>
> We share the reviewer’s concern that, in principle, null-space directions are finite and could be saturated under extreme lifelong learning. However, in our current experimental regimes we do not observe such saturation. On both the heterogeneous MTIL benchmark and the 50-step CIFAR-100 class-incremental setting, the effective rank and null-space ratio of the backbone remain nearly unchanged over long task sequences (see Table A); the remaining null-space dimension stays comfortably large, and performance does not exhibit signs of collapse or instability. This suggests that, for modern over-parameterized backbones and realistic task horizons, the available tail subspace is not quickly exhausted.
>
>   **Table A.** Rank $r_{95}$ and null ratio change across 50 updates in the CIL setting.
>
>   | **Encoder** | **Layer** | **Rank (r95) (Task 0 → 49)** | **Null Ratio (Task 0 → 49)** |
>   | ----------- | --------- | ----------------------------- | ----------------------------- |
>   | **Text**    | $W_k$     | 284.8 → 285.0     | 44.37% → 44.34%   |
>   | **Text**    | $W_q$     | 279.7 → 279.7     | 45.38% → 45.38%   |
>   | **Vision**  | $W_k$     | 358.4 → 358.5     | 53.33% → 53.32%   |
>   | **Vision**  | $W_q$     | 354.1 → 354.2     | 53.90% → 53.88%   |
>
>   *(Note: $r_{95}$ denotes the rank preserving 95% of the energy.)*
>
> While our current experiments do not indicate saturation or instability, in a deployed system one could still monitor simple spectral indicators such as the null ratio or the maximum singular value within the tail subspace, and trigger a lightweight remedy when they exceed a threshold, for example by recomputing the SVD with a stricter energy cutoff or freezing a subset of saturated layers.

---

> ### Author Response · Authors · 2025-11-21
> **Response to Reviewer TeKM (2/2)**
>
> To further address W3 on scalability and potential saturation, we outline how NuSA-CL can be combined with occasional capacity expansion when needed.
>
> If, after many tasks, the remaining null-space dimension were to become too small to effectively learn new tasks, a natural lifecycle would be to periodically distill the continually learned latest model into a backbone with higher capacity (for example, a wider variant of the same architecture) and then restart NuSA-CL on top of this refreshed model. This preserves the accumulated knowledge while restoring sufficient null-space for future tasks.
>
> Conceptually, this yields a “use-then-expand” schedule for model capacity: NuSA-CL first maximizes the utilization of a fixed parameter budget, and only when that budget is genuinely saturated does one increase capacity. In contrast, storage-based methods expand memory and parameters continuously as tasks accumulate, through replay buffers or task-specific modules whose size generally grows with the number of tasks.
>
> From this perspective, NuSA-CL offers a complementary scalability advantage: it is better suited to long-horizon, memory-constrained deployments where continuously growing replay buffers or task-wise modules are undesirable, while still allowing optional capacity expansion when truly needed.
>
> ---
>
> ### **3. On simplicity of the null-space constraint (W2)**
>
> While the update rule is intentionally simple, our experiments show that the null-space constraint is crucial for learning new tasks while preserving past knowledge:
>
> - Across MTIL, NuSA-CL consistently outperforms vanilla LoRA and MiLoRA under the same rank and backbone, while using even fewer trainable parameters than both. This suggests that the improvement does not come from low-rank adaptation alone: NuSA-CL enforces a **persistent** null-space constraint by freezing the subspace bases and training only the intermediate coefficients, and selects this subspace via an adaptive cutoff threshold, rather than using a one-off initialization or fixed rank. In ablations where we relax this constraint (e.g., by unfreezing the bases), performance consistently degrades, indicating that the enforced null-space constraint is crucial for mitigating forgetting.
> - Ablations over subspace choices further show that null-like tail directions yield the lowest forgetting, while principal (Top) directions offer slightly higher single-task Target accuracy at the cost of substantially higher forgetting. This shows that the explicit null-space constraint provides a beneficial stability–plasticity trade-off.
>
> Taken together, these results support our view that NuSA-CL improves effective expressivity under a fixed parameter and memory budget by activating underutilized capacity in directions that minimally interfere with past knowledge, while maintaining competitive performance against storage-based baselines in realistic continual-learning regimes.

---

### Official Review · Reviewer_SL2R · 2025-10-29

**Soundness:** 2
**Presentation:** 2
**Contribution:** 2
**Rating:** 6
**Confidence:** 3

**Summary:**

NuSA-CL addresses continual learning (CL) by balancing stability (retaining old knowledge) and plasticity (adapting to new tasks). The core idea is a low-rank adaptation strategy that constrains task-specific weight updates to the model’s approximate null-space, minimizing interference with previously learned knowledge and preserving the model’s zero-shot learning ability. Unlike methods that rely on replay buffers or high-cost distillation, NuSA-CL requires minimal computation and memory, making it suitable for resource-constrained scenarios. Experimental results demonstrate that the framework not only maintains strong zero-shot transfer performance but also achieves competitive results on continual learning benchmarks.

**Strengths:**

1. Introduces the novel idea of updating in null-space directions (Tail), which is conceptually different from previous approaches that focus on Top singular directions or random subspaces. Combines multimodal adaptation and rank-limited updates, which is a creative integration addressing the stability–plasticity trade-off.
2. Significance Tackles catastrophic forgetting, a central challenge in continual learning. Demonstrates practical low-cost, robust implementation suitable for large-scale models and multimodal tasks.
3. The paper is well-structured, with clear motivation, methodology, and empirical validation. Extensive ablation studies and visualizations (e.g., subspace and rank analysis) make the mechanisms and design choices transparent.
4. Strong experimental evaluation, including comparisons with Top/Random subspaces and analysis of rank choices. Shows robustness to hyperparameters and demonstrates practical feasibility with low SVD overhead.

**Weaknesses:**

1. The method assumes that tasks have sufficiently distinct distributions, enabling interference reduction via orthogonalization or projection. However, when tasks are highly correlated or share overlapping features, the model may struggle to separate old and new knowledge, leading to reduced stability and adaptability.
2. The experimental evaluation is limited to standard vision benchmarks (e.g., CIFAR, ImageNet subsets) with relatively few tasks, focusing only on class-incremental learning. The absence of experiments on more complex multimodal settings such as VQA or image-text retrieval limits the demonstration of generalizability.
3. The method’s performance is sensitive to key hyperparameters, including projection dimension, update ratio, and orthogonalization strength. The lack of an adaptive tuning mechanism may hinder robustness and scalability in real-world, dynamic scenarios.
4. In class-incremental tasks, the method has not been compared with other Parameter-Efficient Fine-Tuning (PEFT) approaches such as LPI, TAM, MoELora，or DIKI, limiting the completeness of its performance validation.

**Questions:**

1. Q: In class-incremental tasks, NuSACL has not been compared with other PEFT approaches such as LPI, TAM, MoELora, or DIKI. How does it perform relative to these methods in terms of both accuracy and memory efficiency?
2. Q: While NuSACL improves stability via orthogonal projection, catastrophic forgetting may still occur in challenging incremental protocols (e.g., many classes, highly overlapping features). Have the authors quantified forgetting under these extreme settings?
3. Q: Although the method is described as lightweight, how does it scale to large models (e.g., ViT-L/14 or multimodal transformers) where repeated SVD computations might be expensive?

---

> ### Author Response · Authors · 2025-11-21
> **Response to Reviewer SL2R (1/2)**
>
> We appreciate the Reviewer SL2R’s constructive feedback and acknowledgment of the novelty of our null-space strategy. Below we clarify our baseline selection, benchmark setup, and scope, and address the raised concerns.
>
> ### **1. Clarification on “Missing” PEFT Baselines (W4 & Q1)**
>
> The reviewer notes that we did not compare NuSA-CL with PEFT approaches such as DIKI, MoELoRA, TAM-CL, or LPI in class-incremental settings [1–4]. We agree that situating NuSA-CL among recent PEFT methods is important and clarify how these approaches relate to our existing baselines.
>
> **(1) Main PEFT comparison (including DIKI) is conducted on MTIL**
>
> Our primary benchmark is MTIL (Multi-Task Incremental Learning), where we explicitly evaluate NuSA-CL against strong PEFT baselines such as DIKI, InfLoRA, and MoE-Adapters. As shown in Table 1, DIKI [1] is a strong parameter-efficient continual tuning method that requires task-specific residual branches and feature statistics. NuSA-CL is competitive while remaining storage-free (no replay buffer or task-wise statistics), and in our MTIL implementation it also achieves **over 3× shorter wall-clock training time** than DIKI. We additionally use Class-Incremental Learning (CIL) as a complementary long-horizon stress test (Table 4), where we focus on conventional CL baselines, while the detailed PEFT comparison is conducted on MTIL.
>
> **(2) MoELoRA / MoE-based adapters, and TAM-CL / replay-based methods**
>
> MoELoRA belongs to the same architectural family as the MoE-Adapters baseline we include: both expand capacity via low-rank experts and a routing/gating mechanism. Although we do not list MoELoRA by name, its core idea is already represented in our MoE-Adapters baseline, against which NuSA-CL is competitive in accuracy and substantially more efficient in memory. TAM-CL combines task-attentive expansion, distillation, and a replay buffer. In our CIL experiments, we compare against representative replay-based CL (iCaRL) and a strong distillation-based CL baseline (ZSCL). NuSA-CL is competitive with these methods while not storing any past data.
>
> Due to time and resource constraints, we could not additionally reproduce MoELoRA and TAM-CL in our experiments. However, since their key design patterns (MoE-style expansion and replay/distillation) are already covered by our baselines, we believe the qualitative trade-offs of these families are reasonably represented in our current comparisons. On MTIL overall, NuSA-CL achieves competitive or better performance while remaining strictly storage-free and using significantly fewer additional parameters and computational resources than the PEFT families suggested by the reviewer. We will clarify this limitation in the revised version.
>
> **(3) LPI in retrieval vs. our classification setting**
>
> LPI is designed for continual vision–language retrieval, whereas our experiments focus on classification-oriented continual learning and on preserving the zero-shot capabilities of the model. The training objectives and evaluation protocols differ substantially. We therefore view LPI as complementary rather than a direct baseline for our setting.
>
> [1] (DIKI) Mind the Interference: Retaining Pre-trained Knowledge in Parameter Efficient Continual Learning of Vision-Language Models. ECCV 2024.
> [2] (MoELoRA) MoELoRA: Contrastive Learning Guided Mixture of Experts on Parameter-Efficient Fine-Tuning for Large Language Models. CoRR 2024.
> [3] (TAM-CL) Task-Attentive Transformer Architecture for Continual Learning of Vision-and-Language Tasks Using Knowledge Distillation. Findings of EMNLP 2023.
> [4] (LPI) Low-rank Prompt Interaction for Continual Vision-Language Retrieval. ACM MM 2024.
>
> ---
>
> ### **2. Benchmark Difficulty and Scope (W2)**
>
> The reviewer remarks that our evaluation is limited to “standard vision benchmarks with relatively few tasks, focusing only on class-incremental learning”. We would like to emphasize that MTIL is a challenging and diverse benchmark specifically designed to probe the stability–plasticity trade-off of vision-language models beyond sequentially learning classes in one dataset:
> - It contains 11 heterogeneous tasks spanning aircraft recognition (FGVC-Aircraft), generic object classification (Caltech101, CIFAR100), textures (DTD), remote sensing (EuroSAT), and other distinct domains (flowers, food, digits, pets, cars, scenes).
> - Several of these domains (e.g., satellite imagery, textures, fine-grained aircraft) are far from CLIP’s pre-training distribution, making them strong tests of plasticity and out-of-distribution generalization.
>
> We agree that extending NuSA-CL to multimodal generative tasks (e.g., VQA, retrieval, continual instruction tuning) is an interesting next step. Architecturally, NuSA-CL is compatible with such models because it only requires access to their weight matrices, and we will briefly discuss this extension in the revised version.

---

> ### Author Response · Authors · 2025-11-21
> **Response to Reviewer SL2R (2/2)**
>
> ### **3. Why Null-Space Updates Remain Stable Under Correlated Tasks (W1 & Q2)**
>
> The reviewer expresses concern that orthogonal projection might fail when tasks are highly correlated and share overlapping features. We agree that this is an important scenario and clarify how NuSA-CL behaves in such settings.
> NuSA-CL identifies directions that the current backbone already uses heavily, as captured by the principal (high-energy) singular vectors of the model weights, and separates them from the remaining low-energy tail directions. Our update rule, $W_t = W_{t-1} + \Delta W_t$, is expressly designed to preserve this principal capacity rather than overwrite it.
>
> For previously learned tasks, the model has already encoded useful features in the principal subspace. For a new task, we decompose its update directions into components that lie within this principal subspace and components that lie in the null / tail subspace, and we constrain learning to the latter. When tasks are highly correlated, their useful directions naturally lie mostly in the existing principal subspace. In this case, NuSA-CL tends to reuse previously learned bases, introduce only small tail updates, and thus maintain stability.
>
> Empirically, this is confirmed on the **50-step CIFAR-100** class-incremental setting (Table 4), where all steps are constructed from a single dataset and thus share highly correlated visual features. NuSA-CL maintains competitive “Last” accuracy over a long sequence, and our spectral analysis shows that the effective rank and null-space ratio remain essentially stable over 50 tasks. This indicates that even under strong feature overlap, null-space guided updates do not lead to collapse or uncontrolled drift: NuSA-CL maintains an average forgetting of only ≈9.9% after all 50 tasks, with early tasks still retaining high accuracy (e.g., Task 0: 98.0% → 87.0% at the end of training).
>
>
> ---
>
> ### **4. Robustness and Number of Hyperparameters (W3)**
>
> The reviewer notes that NuSA-CL may be sensitive to hyperparameters, mentioning “projection dimension, update ratio, and orthogonalization strength.” We fully agree that CL methods should be robust, and we highlight two points.
>
> 1. **Few effective “decision” hyperparameters.**
>
>    In our implementation, NuSA-CL is governed by a small number of global hyperparameters:
>    - a cutoff threshold (e.g., 95% or 99%), which determines how much spectral energy is retained in the principal space and how much remains as null/tail space, and
>    - a maximum update rank $r_{\max}$, which prevents degenerate cases with excessively large ranks.
>
>    We do not introduce separate per-task knobs for “update ratio” or “orthogonalization strength”; these are implicitly determined by the spectral decomposition of the backbone and the chosen cutoff. All experiments use the same global configuration per benchmark.
>
> 2. **Empirical robustness.**
>
>    In Table 3b and Figure 3b, we show that NuSA-CL’s performance varies smoothly across a wide range of ranks and thresholds: both “Avg” and “Last” accuracy remain competitive across different cutoffs, and zero-shot performance is preserved even with conservative rank allocations. By contrast, several baselines we compare to require more extensive hyperparameter tuning, such as per-task learning rates or expert/router configurations.
>
> ---
>
> ### **5. Scalability to Large Models (Q3)**
>
> In our implementation, the SVD is computed once per task and per layer on the weight matrices (e.g., $W_q$, $W_k$), not per batch or per optimization step. For our CLIP ViT-B/16 backbone, these matrices are at most 512×512 in the text encoder and 768×768 in the vision encoder; even for a larger CLIP ViT-L/14–style backbone, the corresponding dimensions are on the order of 768–1024. In practice, the cost of these SVDs is small compared to running full forward–backward passes over the dataset, and we observe only modest wall-clock overhead relative to standard fine-tuning.
>
> Moreover, the spectral step scales with the layer dimensionality and the number of layers, and linearly with the number of tasks, but does not depend on the dataset size or the number of training iterations per task. If computation becomes a concern for very large models or extremely long task sequences, one can reduce the SVD frequency (e.g., refresh the principal/null subspace every $K$ tasks) or use truncated/approximate SVD on selected layers. These engineering choices are orthogonal to the core method; the main user-facing knob remains the energy threshold that trades off plasticity (larger null space) and stability (larger principal space).
>
> We hope these clarifications resolve the reviewer’s concerns and more clearly position NuSA-CL as a storage-free, PEFT-style approach that is competitive with strong replay-, prompt-, and MoE-based methods under realistic memory constraints.

---

### Official Review · Reviewer_4o99 · 2025-10-29

**Soundness:** 3
**Presentation:** 4
**Contribution:** 3
**Rating:** 6
**Confidence:** 4

**Summary:**

The paper proposes NuSA-CL, a memory-free continual learning method for VLMs (e.g., CLIP). It identifies a low-energy “null” subspace via SVD before each task, constrains low-rank updates to this subspace during training, then merges updates into the backbone to avoid parameter growth. Experiments on MTIL (full/5-shot) and CIFAR-100 CIL show strong performance and efficiency; analyses include rank/null-space dynamics, subspace/rank ablations, and core mechanism ablations.

**Strengths:**

1. Simple, natural idea: SVD-based separation of principal vs. null-like directions; persistent constraint prevents drift and reduces interference.
2. Truly memory-free and fixed-size: no replay, no task-specific modules; efficient and scalable.
3. Strong empirical results with comprehensive cost reporting; SOTA among storage-free baselines and close to storage-based methods.
4. Clear, informative ablations (Top/Tail/Random; rmax; persistent constraint; multimodal adaptation) and mechanism evidence (effective-rank, null-ratio trajectories).
5. Practical and robust: negligible SVD overhead; stable across energy thresholds; single-GPU training.

**Weaknesses:**

1. Theory is mostly motivational; tighter links between parameter-space bounds and function-level forgetting would help.
2. Long-horizon spectral drift and null-space quality: Although the method re-computes SVD per task, after many merges the spectrum will evolve. A more systematic study of whether low-energy subspaces gradually become “contaminated” (especially for highly correlated task sequences) and how to monitor/remedy this (e.g., periodic re-orthogonalization, spectral gating) would be valuable.
3. Merged updates limit reversibility; lightweight selective rollback is not explored.
4. Scalability guidance for larger backbones (ViT-L/H) would aid practitioners.

**Questions:**

See weaknesses.

---

> ### Author Response · Authors · 2025-11-21
> **Response to Reviewer 4o99 (1/2)**
>
> We sincerely thank Reviewer 4o99 for the insightful and constructive assessment. We are encouraged by your recognition of NuSA-CL as a "simple, natural idea" that is "truly memory-free," and by your appreciation of its "strong empirical results" and "practical robustness."
> We have carefully considered your feedback regarding the theoretical scope, long-term spectral dynamics, reversibility, and scalability. We believe your comments have helped us significantly strengthen the manuscript. Below, we address your concerns point-by-point.
>
> ### **1. On the theoretical bound and its relation to forgetting (W1)**
> We appreciate the feedback and agree that our current analysis provides a parameter-space bound rather than a function-level guarantee. Our intention is to use this result as a local stability condition: it formalizes how much the parameters are allowed to move along directions that are important for past tasks when applying null-space–constrained updates.
> Intuitively, under standard smoothness assumptions (e.g., bounded sensitivity of the loss with respect to the weights), keeping parameter changes small in those principal directions leads to small local changes in the corresponding predictions. Our theorem captures the parameter-side part of this story by showing that NuSA-CL explicitly limits updates in the principal subspace and pushes most of the adaptation into the approximate null space, thereby controlling interference in the directions that matter most for past tasks.
> The practical relevance of this bound is validated empirically: across our experiments, null-space updates consistently yield lower forgetting (higher Last accuracy) than alternative subspace choices and baselines. We view developing a tighter, more direct theoretical link between this spectral bound and task-level forgetting as an important direction for future work, and we will clarify this positioning in the revised paper.
>
> ---
>
> ### 2. **Long-horizon spectral drift and null-space quality (W2)**
> We wish to clarify that our update rule, $W_t = W_{t-1} + \Delta W_t$, is expressly designed to **preserve previously used principal directions** rather than overwrite them. Even when tasks are highly correlated, the algorithm preferentially allocates updates in the low-energy tail subspace instead of repeatedly modifying high-energy principal directions.
> - **(1) Heterogeneous Tasks (MTIL).**
>   As shown in the paper, NuSA-CL exhibits a structured spectral evolution: the effective rank increases moderately while the null-space ratio decreases only slowly. This suggests gradual capacity reallocation and accumulation of information across tasks, without signs of sudden saturation or collapse.
> - **(2) Stability under Correlated Tasks (CIFAR-100).**
>   To address your specific concern about “contamination,” we additionally analyzed the spectral dynamics in the **50-step CIFAR-100** class-incremental setting, where all steps are constructed from a single dataset and thus share highly correlated visual features. As summarized in the table below, even after 50 updates, the spectrum remains remarkably stable:
>
>   **Table A.** Rank $r_{95}$ and null ratio change across 50 updates in the CIL setting.
>
>   | **Encoder** | **Layer** | **Rank (r95) (Task 0 → 49)** | **Null Ratio (Task 0 → 49)** |
>   | ----------- | --------- | ----------------------------- | ----------------------------- |
>   | **Text**    | $W_k$     | 284.8 → 285.0     | 44.37% → 44.34%   |
>   | **Text**    | $W_q$     | 279.7 → 279.7     | 45.38% → 45.38%   |
>   | **Vision**  | $W_k$     | 358.4 → 358.5     | 53.33% → 53.32%   |
>   | **Vision**  | $W_q$     | 354.1 → 354.2     | 53.90% → 53.88%   |
>
>   *(Note: $r_{95}$ denotes the rank preserving 95% of the energy.)*
>
> While our current experiments do not indicate such instability, in a deployed system one could still monitor simple spectral indicators such as the null ratio or the maximum singular value within the tail subspace, and trigger a lightweight remedy when they exceed a threshold, for example by recomputing the SVD with a stricter energy cutoff or freezing a subset of saturated layers.
>
> If, after many tasks, the remaining null-space dimension were to become too small to effectively learn new tasks, a natural lifecycle would be to periodically distill the continually learned latest model into a backbone with higher capacity (for example, a wider variant of the same architecture) and then restart NuSA-CL on top of this refreshed model. This preserves the accumulated knowledge while restoring sufficient null-space for future tasks. Conceptually, this yields a “use-then-expand” schedule for model capacity: NuSA-CL first maximizes the utilization of a fixed parameter budget, and only when that budget is genuinely saturated does one increase capacity. This stands in contrast to approaches that grow task-specific modules or memory at every step and can be more efficient in long-horizon continual learning deployments.

---

> ### Author Response · Authors · 2025-11-21
> **Response to Reviewer 4o99 (2/2)**
>
> ### 3. On reversibility and selective rollback (W3)
>
> We view "reversibility" as closely related to **machine unlearning**, which is distinct from our primary goal of memory-free continual learning (i.e., accumulation). Modular approaches can indeed support exact rollback by discarding modules, but this comes at the cost of growing storage and architectural complexity. By contrast, NuSA-CL targets a complementary operating point: a **strictly fixed-size, memory-free model** that maintains a single merged backbone, prioritizing deployment simplicity and bounded memory usage over exact rollback.
>
> We agree that lightweight rollback (e.g., via approximate inverse updates or small ring buffers of recent low-rank adapters) is a promising extension, and we will explicitly discuss this trade-off and such potential future directions in the Limitations section.
>
> ---
> ### 4. Practical guidance and scalability to larger backbones (W4)
> We appreciate the constructive suggestion to provide a “practitioner’s guide” for scaling NuSA-CL.
> In our implementation, the SVD is computed **once per task and per layer** on the weight matrices (e.g., $W_q$, $W_k$), not per batch or per optimization step. For our CLIP ViT-B/16 backbone, these matrices are at most **512×512** in the text encoder and **768×768** in the vision encoder; for a larger CLIP ViT-L/14–style backbone, the corresponding matrices are **768×768** (text) and **1024×1024** (vision). In practice, the cost of these SVDs is small compared to running full forward–backward passes over the dataset, and we observe only modest wall-clock overhead relative to standard fine-tuning.
>
> The key hyperparameter is the **cutoff threshold** for rank selection. In our experiments, a single threshold per backbone works robustly across tasks. Practitioners can treat this as a simple knob to balance plasticity (larger null space) and stability (larger principal space), and we will make this guidance more explicit in the revised version.
> If desired, further engineering optimizations are straightforward: e.g., reducing the SVD frequency (refreshing the principal/null subspace every $K$ tasks) or using truncated/approximate SVD on selected layers. These choices are orthogonal to the core method and mainly affect the computational footprint rather than the conceptual framework.
>
> We hope these clarifications address your concerns, and we sincerely appreciate your thoughtful feedback on NuSA-CL.

---

### Official Review · Reviewer_JRHi · 2025-11-01

**Soundness:** 3
**Presentation:** 3
**Contribution:** 2
**Rating:** 6
**Confidence:** 4

**Summary:**

This work proposes NuSA-CL, a lightweight memory-free continual learning framework to address distributional shifts/novel tasks during real-world CLIP usage. NuSA-CL employs low-rank adaptation and null space constraint via SVD to constrain the update of parameters in a way that has low influence on previous knowledge. The experiments demonstrate its effectiveness among memory-free methods.

**Strengths:**

1. The theoretical motivation is clear and reasonable.
2. The experiments demonstrate a favorable result among memory-free methods. The analysis and discussions are comprehensive, showing the method's practicality and robustness.

**Weaknesses:**

1. Storage-free baseline models are not that strong. Then a question is: using null-space may constrain the model expressivity. Is it still able to trade off by increasing the memory cost to get a stronger performance? Or the null-space also constrain the performance upperbound?

**Questions:**

See weakness.

---

> ### Author Response · Authors · 2025-11-21
> **Response to Reviewer JRHi**
>
> We thank Reviewer JRHi for the constructive feedback and for recognizing the theoretical motivation and strong experimental results of NuSA-CL. We address (i) the competitiveness of storage-free NuSA-CL relative to storage-based methods, and (ii) whether null-space constraints limit model expressivity or the performance upper bound.
>
> ### **Q1. Competitiveness relative to storage-based methods**
> We first clarify that storage-free NuSA-CL remains competitive with, and in some cases surpasses, storage-based methods even under a strictly memory-free regime and a fixed backbone parameter budget.
> In the 5-shot MTIL setting (Table 2), NuSA-CL achieves higher Transfer/Avg./Last accuracies (68.1 / 70.3 / 75.4) than InfLoRA (66.8 / 68.9 / 74.8), despite InfLoRA relying on a gradient projection memory. This shows that, under a PEFT continual-learning setting with a fixed backbone parameter budget (all methods use the same backbone), null-space adaptation alone can match or exceed the benefits of explicit gradient replay.
> On the CIFAR-100 CIL benchmark (Table 4), NuSA-CL also attains higher Last accuracy than ZSCL, even though ZSCL is a full fine-tuning baseline that uses knowledge distillation with reference data, indicating robustness under very long task sequences.
>
> ### **Q2. “Is it possible to trade off by increasing memory cost?”**
> We agree that there is a trade-off once one is allowed to maintain task-wise modules or rehearsal buffers in continual learning. When storage-based methods win in the full-shot MTIL setting, this is largely because they allocate task-wise capacity to isolate knowledge via task-specific adapters, routers, or other parameters, which generally require additional memory that grows with the number of tasks.
> Our goal in this work is therefore not to always outperform replay- or architecture-expansion–based methods with unrestricted memory, but rather to ask:
> > **How competitive can a VLM be when it is not allowed to grow or store data, and must continually reorganize a single fixed parameter budget?**
>
> Within this design, NuSA-CL aims to make more effective use of the existing model capacity, instead of relying on extra modules or buffers. The framework is nonetheless compatible with memory-augmented extensions, for example, attaching a small replay buffer or caching a few recent low-rank components on top of our null-space adapters. But exploring such “NuSA-CL + memory” variants is intentionally left outside the scope of this paper, which focuses on the strictly memory-free setting.
>
> ### **Q3. “Does using the null space constrain the model's expressivity?”**
> We argue that constraining updates to the null space does not materially reduce the effective expressivity in practice; instead, it maximizes useful expressivity under a fixed parameter budget by activating under-utilized capacity while preserving previously important directions.
> - **Utilizing underused directions inside the model**
>   As shown in Figure 2, standard full fine-tuning or vanilla LoRA tends to stay in a “lazy” regime: even after learning multiple tasks, the effective rank of the model weights changes only slightly, indicating that a substantial portion of the parameter space remains under-utilized. In contrast, NuSA-CL systematically fills this low-energy null subspace over time, increasing the effective rank while keeping previously occupied principal directions. In other words, it reallocates unused capacity in directions that minimally interfere with past knowledge, rather than simply adding more parameters.
> - **Subspace trade-off: Target vs. Forgetting.**
>   Table 9 directly quantifies the stability–plasticity trade-off across subspaces. At rank $r = 128$:
>   - Adapting the Top (principal) subspace yields slightly higher Target accuracy (85.50%) than the Tail (null-like) subspace (85.22%),
>   - but at a much larger cost in stability: Top suffers 4.44% forgetting, nearly double that of Tail (2.57%).
>   Thus, the Tail subspace incurs at most a $\sim 0.3\%$ drop in single-task Target accuracy but yields a substantial reduction in catastrophic forgetting. Across ranks, the Tail subspace consistently exhibits the lowest forgetting while maintaining Target accuracy very close to Top and Random.
>
> In continual learning, the relevant “performance upper bound” is not the instantaneous accuracy on the current task, but the best achievable trade-off between plasticity (Target) and stability (Forgetting) under a given parameter and memory budget. From this perspective, while the null-space constraint may slightly narrow the set of reachable functions compared to unconstrained full fine-tuning in principle, our experiments show that it improves the practical stability–plasticity frontier by leveraging underused capacity and achieving lower forgetting than principal-subspace updates, without relying on any external storage.

---

### Author Response · Authors · 2025-12-03
**Final Remarks – Summary of Revisions and Clarifications on NuSA-CL (1/2)**

Dear Area Chair and Reviewers,

We sincerely thank the assigned Area Chair and all reviewers for the time and care invested in evaluating our work. We are grateful for the constructive feedback and for the recognition of the value of our contributions. We have uploaded a revised PDF where all additions and modifications are marked in **blue** for ease of inspection.

The reviews highlighted the clear theoretical motivation behind NuSA-CL (JRHi, TeKM) and the novelty of its null-space adaptation strategy (4o99, SL2R). The strictly memory-free nature of the method (4o99, TeKM) and its practical implementation with minimal overhead (4o99, SL2R) were also positively noted. Finally, the reviewers appreciated the extensive empirical validation demonstrating strong performance among storage-free baselines (JRHi, 4o99, TeKM) and the comprehensive ablation studies (4o99, SL2R).

Below, we summarize the core contribution of NuSA-CL and how we have addressed the main concerns raised during the review process.

---

## Core Contribution of NuSA-CL

NuSA-CL targets continual learning for zero-shot vision–language models under a **strictly memory-free, fixed-capacity** setting. Before each task, it identifies an intrinsic “null” (low-energy) subspace of the model weights via SVD and constrains all low-rank updates to this subspace, then merges them back into the backbone. Thus, NuSA-CL **reorganizes existing capacity** without replay buffers, external memories, or task-wise module expansion.

We emphasize this setting because many real-world deployments of foundation models operate under **tight memory, privacy constraints**: past data cannot be stored indefinitely, gradient histories are expensive to keep, and the model footprint must remain fixed (e.g., on edge devices). In such environments, methods that rely on growing parameter counts or maintaining large replay/gradient buffers are hard to adopt in practice.

From this perspective, our central question is “How competitive can a VLM be when it is not allowed to grow or store data, and must continually reorganize a single fixed parameter budget?”. NuSA-CL is designed precisely for this regime: it makes more effective use of existing capacity via null-space adaptation, while remaining compatible with future “NuSA-CL + memory” extensions (e.g., adding a small replay buffer), which we deliberately leave outside the scope of this work.

---

## 1. Competitiveness vs. Storage-Based Methods (JRHi, TeKM, SL2R)

Reviewers asked how NuSA-CL compares to strong storage-based methods in realistic settings. Storage-based baselines in our experiments fall into two families:
(1) **Module-expansion methods** that grow the architecture with task-specific routers/modules (e.g., MoE-adapters, DIKI), and
(2) **Fixed-backbone methods with auxiliary storage**, which keep the backbone size fixed but rely on reference data/teacher models (ZSCL) or gradient projection memories (InfLoRA).

Module-expansion methods (1) can achieve strong performance, but their parameter count and routing cost grow with the number of tasks, making them less practical under memory and latency constraints. NuSA-CL, by contrast, maintains a single fixed-size backbone and uses no task-wise modules. Under a fixed backbone and zero external storage, it remains competitive with, and in several regimes surpasses, fixed-backbone methods with auxiliary storage (2). For convenience, we summarize our performance with InfLoRA on MTIL:

| Setting        | Method   | Transfer | Avg.  | Last  |
|----------------|----------|----------|-------|-------|
| MTIL full-shot | InfLoRA  | 66.2     | 74.2  | **83.6** |
|                | NuSA-CL  | **68.6** | **75.1** | 82.8  |
| MTIL 5-shot    | InfLoRA  | 66.8     | 68.9  | 74.8  |
|                | NuSA-CL  | **68.1** | **70.3** | **75.4** |

NuSA-CL improves **Transfer** and **Avg.** over InfLoRA in both settings, and additionally improves **Last** in the 5-shot regime, despite InfLoRA leveraging a gradient projection memory. On the **CIFAR-100 CIL** benchmark, NuSA-CL also achieves higher **Last** accuracy than ZSCL. These results support our claim that null-space adaptation can remain highly competitive with strong storage-based methods under realistic resource constraints.

---

## 2. Long-Term Stability and Null-Space Saturation (4o99, TeKM, SL2R)

Reviewers raised concerns that the null space might saturate or drift over long task sequences. To address this, we added **Appendix Table 11**, which tracks spectral statistics over a 50-step CIFAR-100 CIL setting with long sequences and highly overlapping features. The results show that both the effective rank and the null-space ratio remain remarkably stable, suggesting that NuSA-CL **reallocates capacity gradually** rather than rapidly exhausting or corrupting the null subspace.

---

### Author Response · Authors · 2025-12-03
**Final Remarks – Summary of Revisions and Clarifications on NuSA-CL (2/2)**

## 3. Theoretical Scope, Expressivity, and Practicality (4o99, JRHi, SL2R, TeKM)

**Theoretical scope (Reviewer 4o99).**
We clarify that our interference bound is best viewed as a **local stability condition in parameter space**, rather than a global function-level guarantee. Under standard smoothness assumptions, limiting movement along high-energy directions constrains prediction changes on past tasks, and we explicitly present our current analysis as a first step toward a tighter link between spectral bounds and forgetting.

**Expressivity vs. stability (Reviewer JRHi).**
Our subspace ablation (Table 9) shows that Tail (null-like) updates offer a better plasticity–stability trade-off than Top updates: Top yields slightly higher immediate task accuracy but substantially larger forgetting, while Tail preserves prior knowledge with only modest loss in on-task performance. Across MTIL, NuSA-CL also outperforms vanilla LoRA and MiLoRA under the **same LoRA rank and backbone**, and ablations that relax orthogonality or mix principal and null-like directions degrade performance, indicating that where  we place low-rank updates (a dynamically recomputed null space with strict orthogonality) is crucial for improving the stability–plasticity frontier under a fixed parameter and memory budget.

**Reversibility and unlearning (Reviewer 4o99).**
Reviewer 4o99 also raised the question of reversibility, which we view as closely related to machine unlearning and somewhat orthogonal to our primary goal of memory-free accumulation. Modular approaches can support exact rollback by discarding task-specific modules, but at the cost of parameter and storage growth proportional to the number of tasks. NuSA-CL deliberately trades exact reversibility for a **strictly fixed-size, memory-free backbone**.

**Hyperparameter robustness (Reviewer SL2R).**
NuSA-CL is governed by very few effective “decision” hyperparameters: a global spectral cutoff that defines the principal vs. tail split, and a stability bound \(r_{\max}\) that prevents excessively large ranks. We do not introduce per-task knobs for “projection dimension,” “update ratio,” or “orthogonalization strength”; these are implicitly determined by the spectrum and cutoff, and Table 3b and Figure 3 show that Avg/Last accuracy and zero-shot performance vary smoothly across a wide range of ranks and thresholds, indicating reasonable robustness.

**Practicality and scalability (Reviewers SL2R, TeKM).**
Section 6.3 now provides explicit *practical guidance*: SVD is computed once per task and per layer, its cost is negligible compared to training iterations, and a single energy-based cutoff per backbone works robustly across tasks. NuSA-CL scales naturally to larger ViT-style backbones when SVD is amortized at the task level, and we include an efficiency comparison showing that our initialization cost is much smaller than InfLoRA’s while achieving higher Avg. performance.

---

Overall, the reviewers’ feedback has helped us more clearly position NuSA-CL as a simple, scalable, and truly memory-free PEFT-style continual learner that is competitive with strong storage-based approaches under fixed resource budgets. We are grateful for these insights and respectfully ask the Area Chair and reviewers to consider these revisions and clarifications in their final evaluation.

Best regards,
**The Authors**

---

### Meta-Review · Area_Chair_8Yvt · 2025-12-07

**Summary:**

The paper offers a simple and principled null-space adaptation approach for continual learning of zero-shot vision-language models under a fixed-capacity setting. The theoretical motivation and technical contributions are acknowledged by the reviewers, together with solid empirical results provided in the paper.

The initial scores were **6664**, and while the authors responded to the reviewers’ concerns in the rebuttal, no reviewer provided a rebuttal-stage follow-up. After carefully reading the reviews and the authors’ rebuttal, I find that the authors have addressed the major concerns with clarity and sufficient evidence. The rebuttal is detailed and convincing, and the revision has effectively improved the overall quality of the manuscript.

Therefore, I believe the revised version is above the acceptance threshold and recommend acceptance as poster.

**Reviewer Concerns:**

The rebuttal largely addressed Reviewer 4o99’s concerns about long-horizon spectral drift and practicality, Reviewer SL2R’s questions on missing PEFT coverage by clarifying baseline scope and robustness, and Reviewer TeKM’s doubts about competitiveness under strict memory-free constraints.

**Reviewer Scores:**

I think Reviewers JRHi, 4o99, and SL2R would likely keep their ratings (6) after reading the rebuttal, while TeKM might increase the score. Therefore, the final ratings might be **6666** if they had been able to participate fully in the discussion.

---

### Decision · Program_Chairs · 2026-01-26

Accept (Poster)